# Sperm chemotaxis is driven by the slope of the chemoattractant concentration field

Héctor Vicente Ramírez-Gómez[1], Vilma Jimenez Sabinina[2], Martín Velázquez Pérez[1], Carmen Beltran[1], Jorge Carneiro[3], Christopher D Wood[4], Idan Tuval[5,6], Alberto Darszon[1]*, Adán Guerrero[4]*

[1]Departamento de Genética del Desarrollo y Fisiología Molecular, Instituto de Biotecnología, Universidad Nacional Autónoma de México (UNAM), Cuernavaca, Mexico; [2]Cell Biology and Biophysics Unit, European Molecular Biology Laboratory (EMBL), Heidelberg, Germany; [3]Instituto Gulbenkian de Ciência (IGC), Rua da Quinta Grande, Oeiras, Portugal; [4]Laboratorio Nacional de Microscopía Avanzada, Instituto de Biotecnología, Universidad Nacional Autónoma de México (UNAM), Cuernavaca, Mexico; [5]Mediterranean Institute for Advanced Studies, IMEDEA (CSIC-UIB), Esporles, Spain; [6]Department of Physics, University of the Balearic Islands, Palma, Spain

**Abstract** Spermatozoa of marine invertebrates are attracted to their conspecific female gamete by diffusive molecules, called chemoattractants, released from the egg investments in a process known as chemotaxis. The information from the egg chemoattractant concentration field is decoded into intracellular $Ca^{2+}$ concentration ($[Ca^{2+}]_i$) changes that regulate the internal motors that shape the flagellum as it beats. By studying sea urchin species-specific differences in sperm chemoattractant-receptor characteristics we show that receptor density constrains the steepness of the chemoattractant concentration gradient detectable by spermatozoa. Through analyzing different chemoattractant gradient forms, we demonstrate for the first time that *Strongylocentrotus purpuratus* sperm are chemotactic and this response is consistent with frequency entrainment of two coupled physiological oscillators: i) the stimulus function and ii) the $[Ca^{2+}]_i$ changes. We demonstrate that the slope of the chemoattractant gradients provides the coupling force between both oscillators, arising as a fundamental requirement for sperm chemotaxis.

**\*For correspondence:**
darszon@ibt.unam.mx (AD);
adanog@ibt.unam.mx (AG)

**Competing interests:** The authors declare that no competing interests exist.

## Introduction

Broadcast spawning organisms, such as marine invertebrates, release their gametes into open water, where they are often subject to extensive dilution that reduces the probability of gamete encounter (*Lotterhos et al., 2010*). In many marine organisms, female gametes release diffusible molecules that attract homologous spermatozoa (*Lillie, 1913*; *Miller, 1985*; *Suzuki, 1995*), which detect and respond to chemoattractant concentration gradients by swimming toward the gradient source: the egg. Although it was in bracken ferns where sperm chemotaxis was first identified (*Pfeffer, 1884*), sea urchins are currently the best-characterized model system for studying sperm chemotaxis at a molecular level (*Alvarez et al., 2012*; *Cook et al., 1994*; *Darszon et al., 2008*; *Strünker et al., 2015*; *Wood et al., 2015*).

The sea urchin egg is surrounded by an extracellular matrix which contains sperm-activating peptides (SAPs) that modulate sperm motility through altering intracellular $Ca^{2+}$ concentration ($[Ca^{2+}]_i$)

and other signaling intermediates (*Darszon et al., 2008*; *Suzuki, 1995*). The biochemical signals triggered by SAPs guide the sperm trajectory toward the egg.

The decapeptide speract is one of best characterized members of the SAP family due to its powerful stimulating effect on metabolism, permeability and motility in *Strongylocentrotus purpuratus* and *Lytechinus pictus* spermatozoa. The binding of speract to its receptor, located in the flagellar plasma membrane, triggers a train of $[Ca^{2+}]_i$ increases in immobilized spermatozoa of both species (*Wood et al., 2003*). This calcium signal was proposed to regulate the activity of dynein motor proteins in the flagellum, and thus potentially modulate the trajectory of free-swimming spermatozoa (*Brokaw, 1979*; *Mizuno et al., 2017*).

A direct link between $[Ca^{2+}]_i$ signaling and sperm motility was established through the use of optochemical techniques to rapidly, and non-turbulently, expose swimming sea urchin spermatozoa to their conspecific attractant in a well-controlled experimental regime (*Böhmer et al., 2005*; *Wood et al., 2005*). Currently, it is well established that the transient $[Ca^{2+}]_i$ increases triggered by chemoattractants produce a sequence of turns and straight swimming episodes (the 'turn-and-run' response), where each turning event results from a rapid increase in the $[Ca^{2+}]_i$ (*Alvarez et al., 2012*; *Böhmer et al., 2005*; *Shiba et al., 2008*; *Wood et al., 2005*). The turn-and-run response seems to be a general requirement for sperm chemotaxis in sea urchins, however it is not sufficient on its own to produce a chemotactic response (*Guerrero et al., 2010a*; *Strünker et al., 2015*; *Wood et al., 2007*; *Wood et al., 2005*).

In spite of 30 years of research since speract's isolation from *S. purpuratus* oocytes (*Hansbrough and Garbers, 1981*; *Suzuki, 1995*), chemotaxis of *S. purpuratus* sperm in the presence of this peptide has not yet been demonstrated (*Cook et al., 1994*; *Darszon et al., 2008*; *Guerrero et al., 2010a*; *Kaupp, 2012*; *Miller, 1985*; *Wood et al., 2015*). A comparison between individual *L. pictus* and *S. purpuratus* sperm responses to a specific chemoattractant concentration gradient generated by photoactivating caged speract (CS) revealed that only *L. pictus* spermatozoa exhibit chemotaxis under these conditions (*Guerrero et al., 2010a*). In that study, *L. pictus* spermatozoa experience $[Ca^{2+}]_i$ fluctuations and pronounced turns while swimming in descending speract gradients, that result in spermatozoa reorienting their swimming behavior along the positive chemoattractant concentration gradient. In contrast, *S. purpuratus* spermatozoa experience similar trains of $[Ca^{2+}]_i$ fluctuations that in turn drive them to relocate, but with no preference toward the center of the chemoattractant gradient (*Guerrero et al., 2010a*).

In the present work, we investigate boundaries that limit sperm chemotaxis of marine invertebrates. Particularly, we examined whether the chemoattractant concentration gradient must have a minimum steepness to provoke an adequate, chemotactic sperm motility response. Previous studies of chemotactic amoebas crawling up a gradient of cAMP, have shown that the slope of the chemical concentration gradient works as a determinant factor in chemotaxis of this species, where the signal-to-noise relationship of stimulus to the gradient detection mechanism imposes a limit for chemotaxis (*Amselem et al., 2012*). In addition, recent theoretical studies by Kromer and colleagues have shown that, in marine invertebrates, sperm chemotaxis operates efficiently within a boundary defined by the signal-to-noise ratio of detecting ligands within a chemoattractant concentration gradient (*Kromer et al., 2018*).

If certain, this detection limit may have prevented the observation and characterization of chemotactic responses on *S. purpuratus* spermatozoa to date. In this study, we identify the boundaries for detecting chemotactic signals of *S. purpuratus* spermatozoa, and show that sperm chemotaxis arises only when sperm are exposed to much steeper speract concentration gradients than those previously employed by *Guerrero et al. (2010a)*. Furthermore, we examined the coupling between the recruitment of speract molecules during sperm swimming (i.e. stimulus function) and the internal $Ca^{2+}$ oscillator, and demonstrate that sperm chemotaxis arises through coupling of these physiological oscillators.

## Results

### Species-specific differences in chemoattractant-receptor binding rates: chemoattractant sensing is limited by receptor density in *S. purpuratus* spermatozoa

Spermatozoa measure the concentration and the changes in concentration of egg-released chemoattractant during their journey. Cells detect chemoattractant molecules in the extracellular media by integrating chemoattractant-receptor binding events. A spermatozoon moving in a medium where the chemoattractant concentration is isotropic will collect stochastic chemoattractant-receptor binding events with a rate $J$, according to *Equation (1)* (*Figure 1*).

$$J = 4\pi Da\bar{c}\frac{N}{N + \pi a/s} = J_{max}\frac{N}{N + \pi a/s} \tag{1}$$

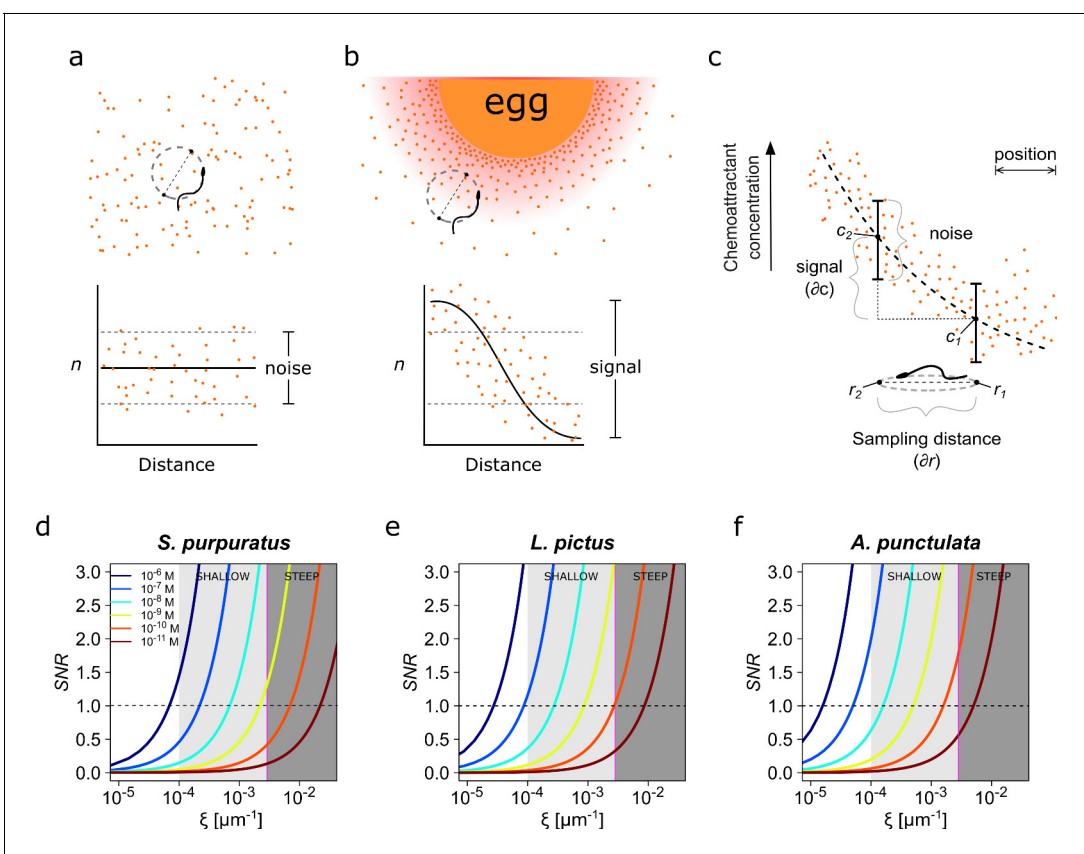

**Figure 1.** Physics of chemoreception. (a) A spermatozoon swimming in an isotropic chemoattractant concentration field, where the number of molecules detected (n) is within the noise of detection. (b) A spermatozoon swimming near to an egg, while chemoattractant molecules are diffusing from its surrounding jelly layer creating a chemoattractant gradient. Note that the signal detected in this case is larger than the detection noise. (c) The assessment of a chemoattractant concentration gradient requires that the signal difference ∂c between two sampled positions ∂r must be greater than the noise. (d–f) The signal-to-noise ratio in the determination of the chemoattractant gradient *SNR* plotted against the relative slope of the chemoattractant concentration gradient in log scale, $\xi = \bar{c}^{-1}\frac{\partial c}{\partial r}$, for different chemoattractant concentrations of speract for either *S. purpuratus* (d), or *L. pictus* (e) spermatozoa, and of resact for *A. punctulata* (f) spermatozoa (see *Supplementary file 1* for the list of parameter values taken in consideration for panels d–f). *S. purpuratus* spermatozoa have lower capacity of detection for the same chemoattractant concentrations at a given ξ than *L. pictus* and *A. punctulata*. The tone of the shaded areas indicates shallow or steep gradient conditions. The horizontal dotted line represents *SNR* = 1; the vertical magenta line represents $\xi = 2.6 \times 10^{-3}$ μm$^{-1}$. Colors of the line traces (from blue to brown) indicate distinct chemoattractant concentrations in the range [$10^{-6}$ – $10^{-11}$ M].

The online version of this article includes the following figure supplement(s) for figure 1:

**Figure supplement 1.** Chemoattractant diffusive currents have a non-linear relationship to receptor coverage.

Where $D$ is the diffusion coefficient of the chemoattractant, $a$ is the radius of the cell, $\bar{c}$ is the mean chemoattractant concentration, $N$ is the number of receptor molecules on the cell surface, $s$ is the effective radius of the chemoattractant molecule, $J_{max}$ is the maximal flux that the cell can experience, and $\frac{N}{N + \pi a/s}$ is the probability that a molecule that has collided with the cell will find a receptor (*Berg and Purcell, 1977*). The quantity $\pi a/s$ is the number of receptors that allows half maximal binding rate for any concentration of chemoattractant, which is hereafter denoted as $N_{1/2}$ (see 1.1. On the estimate of maximal chemoattractant absorption in Appendix 1).

The expression above was used by *Berg and Purcell (1977)* to conclude that the chemoattractant binding and absorption rate saturate as a function of the density of receptors, becoming diffusion limited, that is when $N \gg N_{1/2} = \pi a/s$ the chemoattractant absorption flux becomes $J \cong J_{max}$ (see 1.1. On the estimate of maximal chemoattractant absorption in Appendix 1). If the density of the chemoattractant receptor is such that spermatozoa of the different species operate under this saturated or perfect absorber regime, then any postulated species-specific differences would have to be downstream.

In *Supplementary file 1* we list the biophysical parameters considered for calculating the species-specific rate of binding as a function of the chemoattractant concentration. The different functions of the receptor density and the species receptor density are depicted in *Figure 1—figure supplement 1*. Our calculations (see Appendix 1, section 1.1. On the estimate of maximal chemoattractant absorption) indicate that only *S. purpuratus* spermatozoa operate in a regime for which the rate of chemoattractant uptake is limited by receptor density, therefore it cannot be considered as a perfect absorber. The actual number of speract receptors for this species is approximately $2 \times 10^4$ per sperm cell which is fewer than the estimate of $N_{1/2} \sim 3 \times 10^4$ (*Supplementary file 1*). In contrast, *L. pictus* and *A. punctulata* spermatozoa seem to approximate toward operating as perfect absorbers (*Figure 1—figure supplement 1* and *Supplementary file 1*). Both observations hold when considering the cylindrical geometry of the sperm flagellum. A low number of (non-interacting) receptors, sparsely covering the flagellum (i.e. with a large distance between receptors compared to receptor size) entails a non-saturated diffusive flux that, hence, depends on the number of receptors. The cylindrical geometry of the flagellum strengthens the observation that the larger surface area of the cylinder gives a longer average distance between receptors and, hence, offsetting the saturation of the overall diffusive flux to higher receptor number (see section 1.1. On the estimate of maximal chemoattractant absorption in Appendix 1).

In conclusion, there are meaningful species-specific differences in chemoattractant receptor density which could by themselves explain differences in chemotactic behavior.

## Receptor density constrains the chemoattractant concentration gradient detectable by spermatozoa

A functional chemotactic signaling system must remain unresponsive while the cell swims through an isotropic chemoattractant concentration field and must trigger a directional motility response if the cell moves across a concentration gradient (*Figure 1a–c*). This absolute prerequisite of the signaling system defines the minimal quantitative constraints for reliable detection of a gradient and therefore for chemotaxis.

A cell moving along a circular trajectory in an isotropic chemoattractant field (*Figure 1a*) will collect a random number of chemoattractant-receptor binding events during the half revolution time $\Delta \text{t}$, that has a Poisson distribution with mean $J\Delta \text{t}$ and standard deviation $\sqrt{J\Delta \text{t}}$. Because under these conditions there is no spatial positional information to guide the cell, the chemotactic signaling system must be unresponsive to the fluctuations in the number of binding events expected from the Poisson noise.

The chemotactic response should only be triggered when the cell moves into a concentration gradient (*Figure 1b and c*) sufficiently large to drive binding event fluctuations over the interval $\Delta \text{t}$ with an amplitude that supersedes that of the background noise. As derived in the Appendix 1, section 1.2. A condition for detecting a change in the chemoattractant concentration, the reliable detection of a chemoattractant gradient requires the following condition dependent on the maximal concentration difference experienced during half a revolution and on the mean chemoattractant concentration $\bar{c}$:

$$\left(4\pi Da\bar{c}\frac{N}{N+\pi a/s}\Delta t\right)v\Delta t\frac{\partial c}{\partial r}\bar{c}^{-1} \;>\; \sqrt{4\pi Da\bar{c}\frac{N}{N+\pi a/s}\Delta t} \tag{2}$$

Noting that the left-hand side of the condition represents the chemotactic signal and the right-hand side is a measurement of the background noise, *Equation (2)* can be rewritten in terms of signal-to-noise ratio:

$$SNR = v\Delta t^{3/2}\left(4\pi Da\bar{c}\frac{N}{N+\pi a/s}\right)^{1/2}\xi > 1 \tag{3}$$

Where $v$ is the mean linear velocity ($\frac{\Delta r}{\Delta t}$), where $\Delta r$ is the sampling distance or diameter of the swimming circle, and $\xi = \bar{c}^{-1}\frac{\partial c}{\partial r}$ is the relative slope of the chemoattractant concentration gradient. The quantity $\xi$ measures the strength of the stimulus received when sampling a position $r$, relative to the mean concentration $\bar{c}$ (*Figure 1c*). As $\xi$ increases, the strength of the chemotactic signal increases.

*Equation (3)* means that the ability to reliably determine the source of the attractant depends critically on the relative slope of the chemoattractant concentration gradient $\xi$, which must be steep enough to be distinguishable from noise (*Figure 1b and c*, and *Supplementary file 1*; for further explanation see 1.2. A condition for detecting a change in the chemoattractant concentration in Appendix 1).

We modeled the *SNR* corresponding to different gradients, and within a range of mean concentrations of chemoattractant between $10^{-11}$ to $10^{-6}$ M for three sea urchin species: *S. purpuratus*, *L. pictus* and *A. punctulata* (*Figure 1d-f*). For all species studied, at high mean concentrations of chemoattractant ($10^{-8}$ to $10^{-6}$ M), the change in chemoattractant receptor occupancy experienced at two given distinct positions allows reliable assessment of relatively shallow chemical gradients ($\xi \sim [10^{-3},10^{-4}]$ μm$^{-1}$), with *SNR* > 1 for a wide range of $\xi$ (*Figure 1d-f*). However, at low concentrations of chemoattractant (below $10^{-8}$ M), keeping all other parameters equal, stochastic fluctuations begin to mask the signal. In this low-concentration regime, the steepness of the chemoattractant concentration gradient is determinant for chemoattractant detection. Shallow gradients result in insufficient *SNR*, while steeper chemoattractant gradients ($\xi > 10^{-3}$ μm$^{-1}$) are dependably detected by spermatozoa, that is *SNR* > 1 (*Figure 1d-f*).

Previous reports show that *A. punctulata* spermatozoa are very sensitive to resact (presumably reacting to single molecules) due the high density of resact receptors ($\sim 3 \times 10^5$ per cell), which allows them to sense this chemoattractant at low picomolar concentrations (*Kashikar et al., 2012*). In contrast, *L. pictus* and *S. purpuratus* spermatozoa bear lower densities of chemoattractants receptors, approximately $6.3 \times 10^4$ and $2 \times 10^4$ receptors/cell, respectively (*Nishigaki et al., 2001*; *Nishigaki and Darszon, 2000*). According to these species-specific differences in chemoattractant receptor densities, *Figure 1d–f* suggests that the spermatozoa of *A. punctulata* are likely more sensitive to resact, than those of either *L. pictus* or *S. purpuratus* species to the same mean concentration gradients of speract; with the spermatozoa of *S. purpuratus* being less sensitive than those of *L. pictus* species to equivalent speract gradients and mean concentrations. Moreover, the constraints on *SNR* imply that *S. purpuratus* spermatozoa should only respond to the chemoattractants at higher mean speract concentrations and at steeper gradients than those that elicit chemotaxis in *L. pictus* spermatozoa (compare *Figure 1d and e*).

To understand the differential sensitivity between the spermatozoa of *S. purpuratus* and *L. pictus* we analyzed the scenario in which the capacity to detect the gradient for both spermatozoa species were equal, that is they would have the same signal-to-noise ratios, $SNR_{purpuratus} = SNR_{pictus}$. We compute the ratio of the slopes of the speract concentration gradient experienced by either *S. purpuratus* or *L. pictus* spermatozoa, which represents a scaling factor (*SF*) in the gradient slope, expressed as:

$$SF = \frac{\xi_{purpuratus}}{\xi_{pictus}} = \left(\frac{v_{pictus}}{v_{purpuratus}}\right)\left(\frac{\Delta t_{pictus}}{\Delta t_{purpuratus}}\right)^{3/2}\left(\frac{Z_{pictus}}{Z_{purpuratus}}\right)^{1/2} \sim 3 \tag{4}$$

where $Z = \left(\frac{Na}{N + \pi a/s}\right)$ is the probability that a speract molecule that has collided with the cell will bind to a receptor (**Berg and Purcell, 1977**), multiplied by the radius $a$ of the cell.

The estimation of the scaling factor $SF$ predicts that *S. purpuratus* spermatozoa should undergo chemotaxis in a speract gradient three times steeper than the gradient that elicits chemotaxis in *L. pictus* spermatozoa, with $\xi_{purpuratus} \sim 3\xi_{pictus}$.

In summary, the chemoreception model suggests that *S. purpuratus* spermatozoa detect chemo-attractant gradients with lower sensitivity than those of *L. pictus*. It also predicts that *S. purpuratus* spermatozoa may detect chemoattractant gradients in the $10^{-9}$ M regime with sufficient certainty only if the slope of the chemoattractant concentration gradient is greater than $3 \times 10^{-3}$ μm$^{-1}$ (i.e. steep concentration gradients) (**Figure 1d**).

If the latter holds true, then *S. purpuratus* spermatozoa should be able to experience chemotaxis when exposed to steeper speract gradients than those tested experimentally so far. Given this prediction, we designed and implemented an experimental condition for which we expect *S. purpuratus* spermatozoa to experience chemotaxis. In general terms, this scaling rule for sensing chemoattractant gradients might also apply for other species of marine invertebrates.

## *S. purpuratus* spermatozoa accumulate at steep speract concentration gradients

Our experimental setup is designed to generate determined speract concentration gradients by focusing a brief (200 ms) flash of UV light along an optical fiber, through the objective, and into a field of swimming *S. purpuratus* spermatozoa containing caged-speract (CS) at 10 nM in artificial sea water (**Guerrero et al., 2010a**; **Tatsu et al., 2002**). To test experimentally whether *S. purpuratus* spermatozoa undergo chemotaxis, as predicted from the chemoreception model, we varied the slope of the speract concentration gradient by separately employing four optical fibers of distinct diameters, arranged into five different configurations (*f1, f2, f3, f4, f5*) (**Figure 2c**).

Each configuration produces a characteristic pattern of UV illumination within the imaging field (**Figure 2c**). The UV intensity, measured at the back focal plane of the objective for each fiber configuration, is shown in **Supplementary file 2**. The spatial derivative of the imaged UV light profile was computed as a proxy for the slope of the speract concentration gradient (**Figure 2b**). By examining these UV irradiation patterns, the highest concentration of speract released through photo-liberation from CS is generated by the *f5* fiber, followed by *f4>f3>f2>f1* (**Figure 2a**). The steepest UV irradiation gradients are those generated by the *f2, f3* and *f5* fibers (**Figure 2b**).

Irrespective of the optical fiber used, the photo-activation of caged speract triggers the stereotypical Ca$^{2+}$-dependent motility responses of *S. purpuratus* spermatozoa (**Figure 2d**, **Video 1**, **Appendix 1—video 1** and **Appendix 1—videos 4**, **5** and **6**). To determine whether these changes lead to sperm accumulation, we developed an algorithm, which automatically scores the number of spermatozoa at any of the four defined concentric regions (R1, R2, R3, and R4) relative to the center of the speract concentration gradient (**Figure 3a** and **Figure 3—figure supplement 1**).

As you can see in **Table 1 Supplementary file 1**, the photo-liberation of speract through the different fibers used here triggered various response types (**Figure 3b and c** and **Figure 3—figure supplement 2**). Negative controls (Low [Ca$^{2+}$]$_i$ or High extracellular K$^+$ ([K$^+$]$_e$) for *f2* gradient) did not show increased sperm numbers in any region (**Figure 3b** and **Figure 3—figure supplement 2**; **Appendix 1—videos 2** and **3**, respectively).

In summary, *S. purpuratus* spermatozoa accumulate significantly toward the center of the speract gradients generated by the *f2*- and *f3*-fibers (**Figure 3b**), which provide UV light profiles with steeper slopes compared to the *f1* and *f4* fibers (**Figure 2b**). These observations agree with the chemoreception model, in that spermatozoa exposed to steeper gradients experience lower uncertainty (i.e. higher $SNR$) to determine the direction of the source of the chemoattractant.

Notably, the use of fibers *f4* and *f5* uncages higher concentrations of speract (by providing higher UV energies than other fibers) (**Figure 2a** and **Supplementary file 2**), yet they do not trigger the maximum accumulation of *S. purpuratus* spermatozoa at the center of the chemoattractant field.

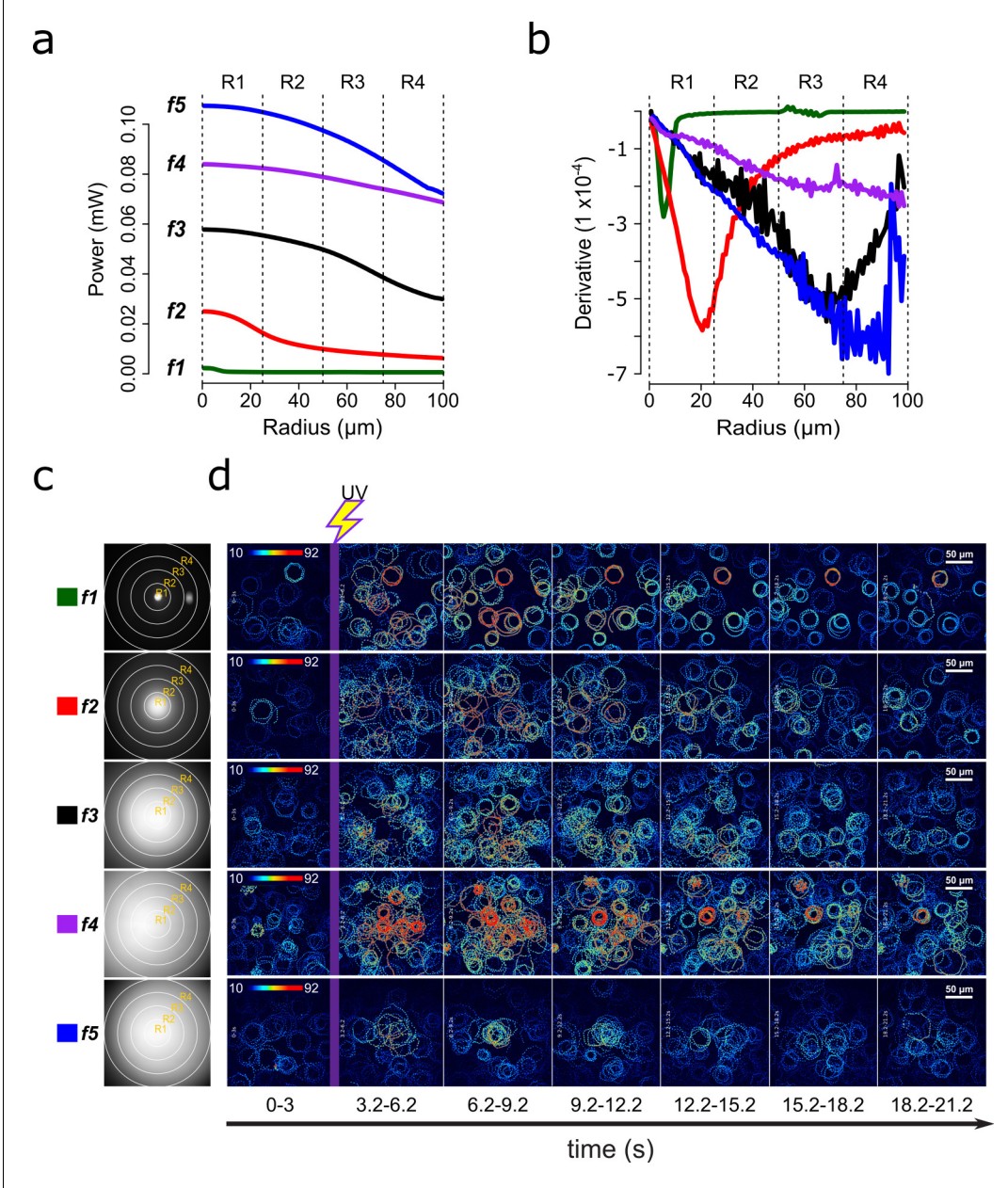

**Figure 2.** Screening of speract concentration gradients. (**a**) Radial profile and its derivative (**b**) of the UV light scattered at the glass-liquid interface for each optical fiber (f1–f5). (**c**) Spatial distribution of the UV flash energy for each fiber. (**d**) Representative motility and $[Ca^{2+}]_i$ responses of *S. purpuratus* spermatozoa exposed to different concentration gradients of speract. $F-F_0$ time projections, showing spermatozoa head fluorescence at 3 s intervals before and after photoactivation of 10 nM caged speract in artificial seawater with 200 ms UV flash. The pseudo-color scale represents the relative fluorescence of fluo-4, a $[Ca^{2+}]_i$ indicator, showing maximum (red) and minimum (blue) relative $[Ca^{2+}]_i$. Scale bars of 50 μm.

## *S. purpuratus* spermatozoa undergo chemotaxis upon exposure to steep speract gradients

The sperm accumulation responses observed in any of *f2* and *f3* conditions suggest that the slope of the chemoattractant concentration gradient might indeed function as a driving force for sperm chemotaxis. However, the accumulation of spermatozoa at the center of the field might also imply other factors, such as cell trapping, or cell death (*Yoshida and Yoshida, 2011*).

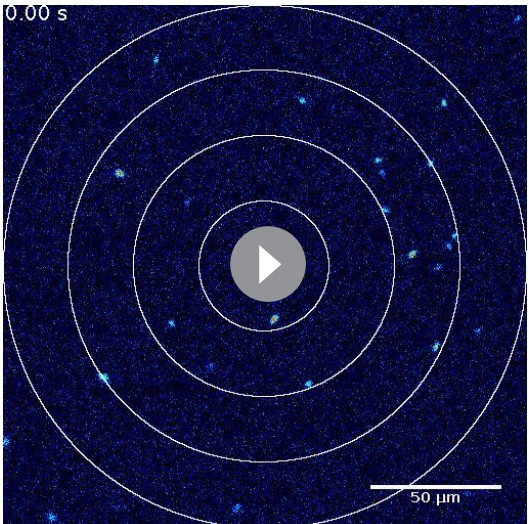

0.00 s

50 µm

**Video 1.** Typical motility and Ca²⁺ responses of *S. purpuratus* spermatozoa toward an *f2*-generated speract concentration gradient. Spermatozoa swimming in artificial sea water containing 10 nM caged speract, 3 s before and 5 s after 200 ms UV irradiation. An optical fiber of 0.6 mm internal diameter (*f2*) was used for the UV light path to generate the speract concentration gradient. Real time: 30.8 frames s⁻¹, 40x/1.3NA oil-immersion objective. Note that spermatozoa located at R2, R3 and R4 regions prior to speract exposure swim up the speract concentration gradient, toward the center of the imaging field. The pseudo-color scale represents the relative fluorescence of fluo-4, a Ca²⁺ indicator, showing maximum (red) and minimum (blue) relative [Ca²⁺]ᵢ. Six *S. purpuratus* spermatozoa were manually tracked for visualization purposes. Scale bar of 50 µm.
https://elifesciences.org/articles/50532#video1

To more reliably scrutinize the trajectories described by *S. purpuratus* spermatozoa in response to speract gradients, chemotactic behavior was quantified using a chemotactic index (CI) that considers the sperm speed and direction both before and after the chemotactic stimulus (see *Figure 4a and b*). This CI takes values from −1 (negative chemotaxis) to 1 (positive chemotaxis), with 0 being no chemotaxis at all (*Video 2*). The temporal evolution of the CI, for each of *f1, f2, f3, f4, f5* speract concentration fields, was computed (*Figure 4c*), and their distributions across time were analyzed by a binomial test (*Figure 4d*, and *Appendix 1—video 7* (for further explanation, see *Chemotactic index* section in Materials and methods).

The speract fields created by fibers *f2, f3* and *f5* produce significantly positive CI values compared to other conditions (*f1, f4* and negative controls), confirming that steeper speract concentration gradients trigger chemotactic responses in *S. purpuratus* spermatozoa. Again, the lack of chemotactic responses in *S. purpuratus* spermatozoa observed by *Guerrero et al. (2010a)*, was reproduced through stimulation with *f4*, zero Ca²⁺, or High K⁺ experimental regimes (a scrutiny of non-chemotactic cells is presented in *Figure 4—figure supplement 1* and section 2.7. Sperm swimming behavior in different chemoattractant gradients in Appendix 1).

Chemotactic efficiency, which in our work is reported by CI, contains information regarding the capability of single cells to detect and undergo a direct response toward a chemotactic stimulus. It also provides information about the percent of responsive cells that, after detecting a stimulus, can experience chemotaxis. As sperm chemotaxis, and chemotaxis in general, has evolved to operate optimally in the presence of noise (*Amselem et al., 2012*; *Kromer et al., 2018*; *Lazova et al., 2011*), we examined the boundary of *SNR* where sperm chemotaxis operates efficiently for *S. purpuratus* spermatozoa (*Figure 4e*). Take into account that in the regime of *SNR* < 1, chemotactic efficiency scales monotonically; for *SNR* > 1, saturation or adaptation mechanisms might impinge on the chemotactic efficiency, as reported in other chemotactic signaling systems (*Amselem et al., 2012*; *Kromer et al., 2018*; *Lazova et al., 2011*). In agreement with these results, we found that the percentage of *S. purpuratus* spermatozoa experiencing relocation increases monotonically with the *SNR* (*Figure 4f*), within the noise limits of 0.1 < *SNR* < 0.8, which is also in agreement with the findings of sperm chemotaxis operating optimally in the presence of noise (*Amselem et al., 2012*; *Kromer et al., 2018*; *Lazova et al., 2011*).

## The magnitude of slope of the gradient is a major determinant of sperm chemotaxis

The spatial derivative of the UV profiles shown in *Figure 2b* indicates that the steeper light gradients generated from UV irradiation are those of *f2, f3* and *f5*, which are assumed to generate the steepest speract gradients of similar form. This assumption is strictly valid at the instant of UV exposure; subsequently the speract gradient dissipates over time with a diffusion coefficient of D ≈ 240 µm²s⁻¹. However, the gradient steepness that each spermatozoon experiences during swimming is

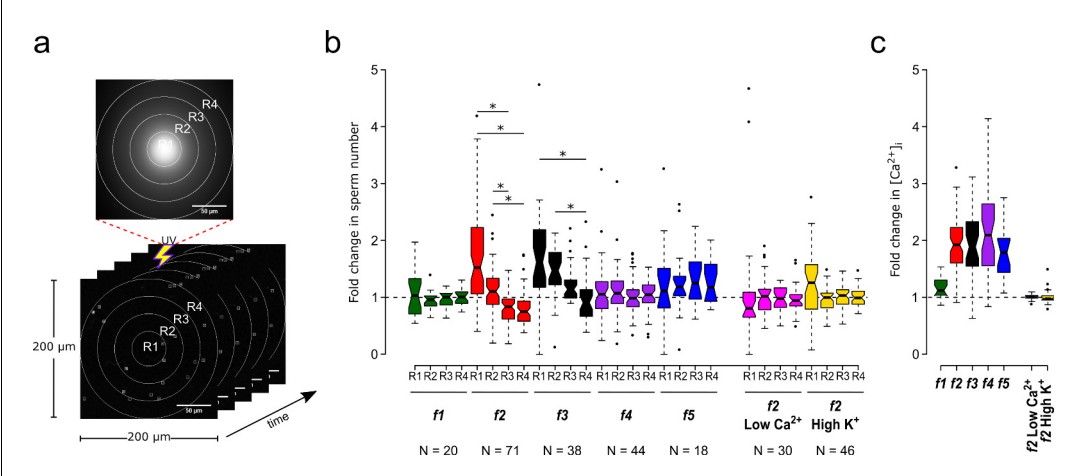

**Figure 3.** Motility and $[Ca^{2+}]_i$ responses of *S. purpuratus* spermatozoa exposed to specific concentration gradients of speract. (**a**) The positions of the sperm heads within the imaging field are automatically assigned to either R1, R2, R3 or R4 concentric regions around the centroid of the UV flash intensity distribution. Each ROI was also used to obtain the sperm head fluorescence from the raw video microscopy images (as the mean value of the ROI) (see *Figure 3—figure supplement 1*). Scale bar of 50 µm. (**b**) Fold change in sperm number, defined as the number of spermatozoa at the peak of the response (6 s) relative to the mean number before speract stimulation (0–3 s) (see *Figure 3—figure supplement 2*). (**c**) Relative changes in $[Ca^{2+}]_i$ experienced by spermatozoa at the peak response (6 s) after speract stimulation. Negative controls for spermatozoa chemotaxis are artificial seawater with nominal $Ca^{2+}$ (Low $Ca^{2+}$); and artificial seawater with 40 mM of $K^+$ (High $K^+$). Both experimental conditions prevent chemotactic responses by inhibiting the $Ca^{2+}$ membrane permeability alterations triggered by speract; the former disrupts the $Ca^{2+}$ electrochemical gradient, and the later disrupt the $K^+$ electrochemical gradient required as electromotive force needed to elevate $pH_i$, and to open $Ca^{2+}$ channels. The central line in each box plot represents the median value, the box denotes the data spread from 25% to 75%, and the whiskers reflect 10–90%. The number of experiments is indicated on the bottom of each experimental condition. We used the same number of experiments for the relative change in $[Ca^{2+}]_i$ (right panel). *Statistical significance, $p<0.05$; multiple comparison test after Kruskal-Wallis.

The online version of this article includes the following figure supplement(s) for figure 3:

**Figure supplement 1.** Automatic segmentation of swimming spermatozoa.

**Figure supplement 2.** Sperm response to speract photo-release, collated data from individual experiments.

**Figure supplement 3.** Spontaneous *vs.* speract-induced $[Ca^{2+}]_i$ oscillations.

determined by the combination of UV flash duration, the speract diffusion time, and the sperm motility response by itself.

In nature, spermatozoa of external fertilizers tend to swim in spiral 3D trajectories. However, under the experimental conditions explored in this research, we analyzed sperm swimming in 2D circular-like trajectories confined at a few microns above the coverslip. The UV flash that sets the initial chemoattractant distribution was focused at the imaging plane (~1–4 µm above the coverslip) (*Nosrati et al., 2015*). Hence, the correct diffusion problem corresponds to that of a 2D diffusing regime. We sought to understand how the stimulus function, which *S. purpuratus* spermatozoa experience during the accumulation of bound speract throughout their trajectory, influences their motility response. For this purpose, we computed the spatio-temporal dynamics of the speract gradient for *f1*, *f2*, *f3*, *f4* and *f5* fibers (*Figure 5a and b* and *Figure 5—figure supplement 1*). and analyzed the trajectories of spermatozoa swimming in these five distinct speract

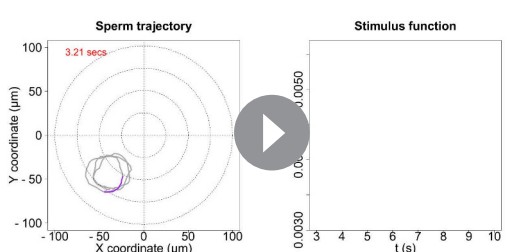

**Video 2.** Sperm trajectory analysis and stimulus function. Single-cell analysis was performed for approximately 1000 sperm trajectories for the different speract gradients (*f1-f5* and negative controls). The sperm trajectory shown here is representative of a chemotactic sperm. This analysis was implemented after speract uncaging at 3 s (from 3.2 to 10 s). Trajectory before, after and during the 200 ms UV flash is shown in gray, black and purple, respectively. https://elifesciences.org/articles/50532#video2

gradient configurations (*Figure 5c*, *Figure 5—figure supplement 2a* and *Figure 5—figure supplement 2c*). Moreover, we examined the stimulus function of individual spermatozoa in response to each of the five speract gradient forms (*Figure 5e*, *Figure 5—figure supplement 2b*, *Figure 5—figure supplement 2d* and *Video 3*).

The model of chemoreception presented in the previous sections (see *Equations (2) and (3)*) predicts a scaling rule for chemotactic responses between *S. purpuratus* and *L. pictus* spermatozoa of $SF \sim 3$ (*Equation (4)*). The derivatives of the UV-irradiation profiles shown in *Figure 2b* indicate that the *f2*, *f3*, and *f5* fibers generate steeper speract gradients than the f1 and *f4* fibers.

To determine the direction of the chemoattractant concentration gradient, the signal difference $\partial c$ between two sampled positions $\partial r$ must be greater than the noise (*Figure 1a*). To test the prediction of the chemoreception model, we computed the local relative slope of the chemoattractant concentration gradient $\xi$ detected by single spermatozoa exposed to a given speract concentration gradient, with $\xi = \bar{c}^{-1} \frac{\partial c}{\partial r}$ (*Figure 5e*).

We found that, in agreement with the chemoreception model, the maximum relative slope of the chemoattractant concentration gradient $\xi_{max} = max(\xi_1, \xi_2, \xi_3, \dots, \xi_n)$ required by *S. purpuratus* spermatozoa to undergo chemotaxis is created when the *f2* and *f3* fibers are employed to generate speract gradients (*Figure 5e*). This relative slope of the chemoattractant concentration gradients is at least three times greater than that experienced when exposed to the *f4*-generated speract gradient (*Figure 6b*). In addition, *L. pictus* spermatozoa undergo chemotaxis when exposed to the *f4* speract gradient, which is 2–3 times smaller than that required by *S. purpuratus* (*Figure 6b*). These findings support the predicted scaling rule for the detection of the speract concentration gradient between *L. pictus* and *S. purpuratus* spermatozoa (*Figure 6b and c*).

## The slope of the speract concentration gradient is the critical determinant for the strength of coupling between the stimulus function and the internal Ca²⁺ oscillator

Friedrich and Jülicher proposed a general theory that captures the essence of sperm navigation traversing periodic paths in a non-homogeneous chemoattractant field, in which the sampling of a stimulus function *S(t)* is translated by intracellular signaling into the periodic modulation of the swimming path curvature *k(t)* (*Friedrich and Jülicher, 2008*; *Friedrich and Jülicher, 2007*; *Riedel et al., 2005*).

As a result, the periodic swimming path drifts in a direction that depends on the internal dynamics of the signaling system. In this theory, the latency of the intracellular signaling (i.e. the $[Ca^{2+}]_i$ signal), expressed as the phase shift between *S(t)* and *k(t)*, is a crucial determinant of the directed looping of the swimming trajectory up the chemical concentration field (*Friedrich and Jülicher, 2009*; *Friedrich and Jülicher, 2008*).

Even though this conceptual framework provides insights into the mechanism governing sperm chemotaxis, it does not explore the scenario where chemoattractants trigger an autonomous $[Ca^{2+}]_i$ oscillator (*Aguilera et al., 2012*; *Espinal et al., 2011*; *Wood et al., 2003*), which suggests that sperm chemotaxis might operate in a dynamical space where two autonomous oscillators, namely the stimulus function and the internal Ca²⁺ oscillator, reach frequency entrainment (*Figure 6a*).

To test the hypothesis that the slope of the speract concentration gradient regulates the coupling between the stimulus function and the internal Ca²⁺ oscillator triggered by speract, we made use of a generic model for coupled phase

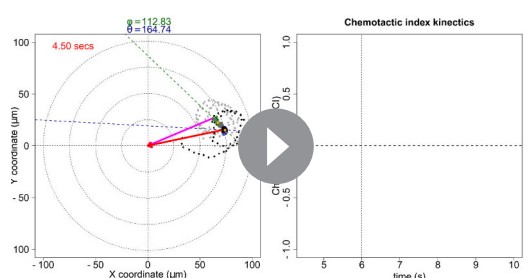

**Video 3.** Sperm trajectory analysis and chemotactic index (CI). Single-cell analysis was performed for approximately 1000 sperm trajectories from the different speract gradients (*f1-f5* and negative controls). Angle ϕ is calculated just once and is always the same for each sperm trajectory. Angle θ is calculated per frame of the video for each sperm trajectory, resulting in the chemotactic index kinetics for each sperm trajectory (right panel). The sperm trajectory shown here represents a chemotactic sperm. This analysis was implemented from 4.5 s to 10 s. Speract uncaging was induced at 3 s. Trajectory before and after speract release is shown in gray and black dots, respectively.
https://elifesciences.org/articles/50532#video3

oscillators (*Pikovsky et al., 2003*). In its simplest form, the model describes two phase oscillators of intrinsic frequencies $\omega_1$ and $\omega_2$ coupled with a strength $\gamma$ through the antisymmetric function of their phase difference $\Phi = \varphi_1 - \varphi_2$. The time evolution of $\Phi$ then follows an Adler equation $d\Phi/dt = \Delta\omega - 2\gamma \, sin(\Phi)$, which is the leading order description for weakly-coupled non-linear oscillators. In the present case, the two coupled oscillators are the internal $Ca^{2+}$ oscillator and the oscillations in the stimulus function induced in spermatozoa swimming across a speract gradient (*Figure 6a*). The former occurs even for immotile cells, for which there are no stimulus oscillations under a spatially uniform speract field (*Figure 6—figure supplement 1*, and *Appendix 1—video 8*); while the latter exists under two tested negative controls: cells swimming in Low $Ca^{2+}$ and in High $K^+$ artificial sea water, both of which inhibit $Ca^{2+}$ oscillations (see *Figure 3c*, *Figure 3—figure supplement 2* and *Appendix 1—videos 2* and *3*, respectively).

Wood et al., showed that immobilized *S. purpuratus* spermatozoa might experience spontaneous $Ca^{2+}$ transients (*Wood et al., 2003*) (see *Figure 6—figure supplement 1*). To provide insight into the mechanism of sperm chemotaxis we characterized and compared the spontaneous vs. the speract-induced $[Ca^{2+}]_i$ oscillations, and conclude that they are of different oscillatory nature, hence the spontaneous oscillations do not have a role in sperm chemotaxis (see *Figure 3—figure supplement 3* and section 2.8. Spontaneous vs. speract-induced $[Ca^{2+}]_i$ oscillations in Appendix 1).

There are two immediate predictions from the Adler model: first, there is a minimum coupling strength necessary for the two oscillators to synchronize ($\gamma_{min} = \Delta\omega/2$). For weaker coupling (i.e. $\gamma < \gamma_{min}$), the two oscillators run with independent frequencies and, hence, the phase difference increases monotonically with time; second, and within the synchronous region (i.e. $\gamma > \gamma_{min}$), the phase difference between the oscillators is constant and does not take any arbitrary value, but rather follows a simple relation to the coupling strength ($\Phi_{sync} = arcsin(\Delta\omega/2\gamma)$). *Figure 6d* shows the two regions in the parameter space given by $\Delta\omega$ and $\gamma$. The boundary between these two regions corresponds to the condition $\gamma = \gamma_{min}$ and it delimits what is known as an Arnold's tongue.

We measured the difference in intrinsic frequency by looking at the instantaneous frequency of the internal $Ca^{2+}$ oscillator just before and after the speract gradient is established. The range of measured $\Delta\omega$ is shown in *Figure 6d* as a band of accessible conditions in our experiments (mean of $\Delta\omega$, black line; mean ± standard deviation, green dashed lines). If the driving coupling force between the oscillators is the maximum slope of the speract concentration gradient, that is $\gamma = \xi_{max}$, we would expect to find a minimum slope ($\hat{\imath} * \overline{max}$) below which no synchrony is observed. This is indeed the case as clearly shown in *Figure 6b, e and f* (magenta line). Moreover, and for cells for which synchronization occurs, the measured phase difference is constrained by the predicted functional form of $\Phi_{sync} = \Phi_{sync}(\Delta\omega, \gamma)$ as can be verified from the collated data shown in *Figure 6e and f* within the theoretical estimates (see also *Figure 6—figure supplement 2*). Altogether, the excellent agreement of this simple model of coupled phase oscillators with our data, points to the slope of the speract concentration gradient as the driving force behind the observed synchronous oscillations and, as a result, for the chemotactic ability of sea urchin spermatozoa.

## Discussion

What are the boundary conditions that limit a sperm's capacity to determine the source of guiding molecules?

During their journey, spermatozoa must measure both the concentration and change on concentration of chemoattractants. Diffusing molecules bind to receptors as discrete packets arriving randomly over time with statistical fluctuations, imposing a limit on detection. By following the differences in the mean concentration of chemoattractants, sampled at a particular time, spermatozoa gather sufficient information to assess the source of the gradient. However, there is a lower detection limit to determine the direction of the chemical gradient, which depends on the swimming speed of the sperm, the sampling time, and as shown in this work, on the steepness of the slope of the chemoattractant concentration gradient.

For almost three decades, chemotaxis had not been observed for the widely-studied *S. purpuratus* species under diverse experimental conditions, raising doubts about their chemotactic capabilities in response to the speract concentration gradients (*Cook et al., 1994*; reviewed in *Guerrero et al. (2010a)*; *Guerrero et al. (2010b)*; *Solzin et al. (2004)*. The observed lack of chemotactic responses by these spermatozoa has been recognized as an 'anomaly' in the field - if we aspire

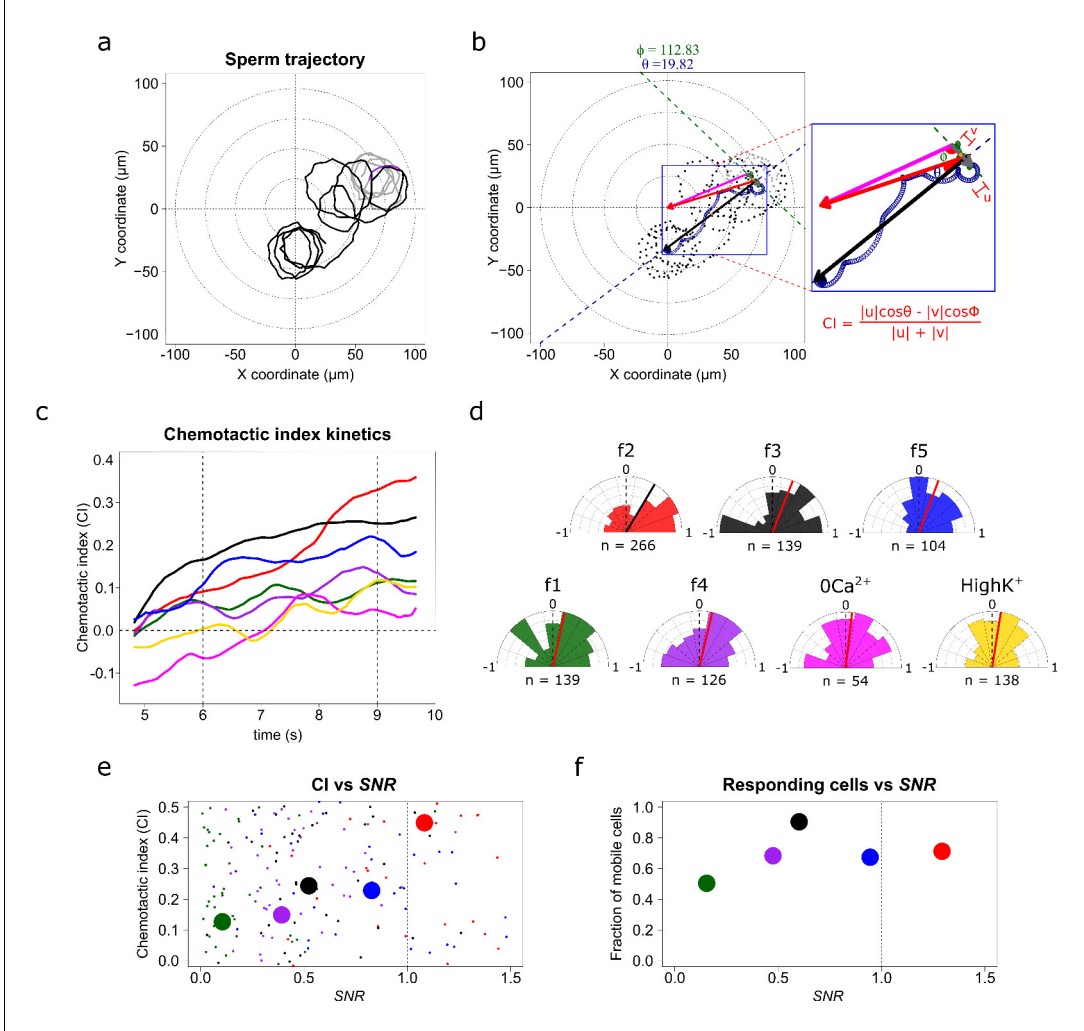

**Figure 4.** *S. purpuratus* spermatozoa selectively experience chemotaxis toward specific speract gradients. (a) Sperm trajectory before (gray) and after (black) the UV irradiation (purple). (b) Definition of a chemotactic index to score chemotactic responses. Dots represent sperm trajectory before (gray) and after (black) UV irradiation. Green and blue empty spirals indicate the smoothed trajectory before and after UV irradiation. Gray and black vectors are the progressive sperm displacement before and after stimulation, respectively; and the v and u vectors are the linear speed before and after stimulation; and ϕ and θ are the angles to their corresponding reference vectors to the center of the imaging field – the highest UV irradiated area, (magenta and red, respectively). Chemotactic index (CI) is defined as in the inset (see also *Video 2*). (c) Temporal evolution of the chemotactic index. Functions were calculated from the median obtained from sperm trajectories of each of *f1, f2, f3, f4, f5, f2-ZeroCa²⁺*, and *f2-HighK⁺* experimental conditions (*Appendix 1—video 7*). (d) Radial histograms of CI computed at second 9 (vertical dotted line at panel **c**). Significant differences (*Binomial test, p-value<0.05*) were observed only for *f2, f3* and *f5* fibers, compared to controls. n represents the number of individual sperm trajectories analyzed. (e) CI as a function of the signal-to-noise ratio (*SNR*). Each parameter was calculated for single cells. Large filled points represent the median for each gradient condition distribution. (f) Fraction of responding cells as a function of the *SNR* (spermatozoa whose effective displacement was above the unstimulated cells). The apparent diffusion of the swimming drifting circle of unstimulated *S. purpuratus* spermatozoa is $D_{app} = 9 \pm 3 \ \mu m^2 \ s^{-1}$ (*Friedrich, 2008*; *Friedrich and Jülicher, 2008*; *Riedel et al., 2005*), here responsive cells were considered by showing a $D_{app} = 9 \ \mu m^2 \ s^{-1}$, and were evaluated at second 9.

The online version of this article includes the following figure supplement(s) for figure 4:

**Figure supplement 1.** Sperm swimming behavior in response to different chemoattractant gradients.

to generalize and interpret findings in sea urchin spermatozoa to chemotactic responses in other systems, then it is critical to accommodate and account for any apparent outliers, and not ignore them as inconveniently incongruent to the model.

To examine whether *S. purpuratus* spermatozoa are able to detect spatial information from specific chemoattractant concentration gradient, we use a model of chemoreception developed by

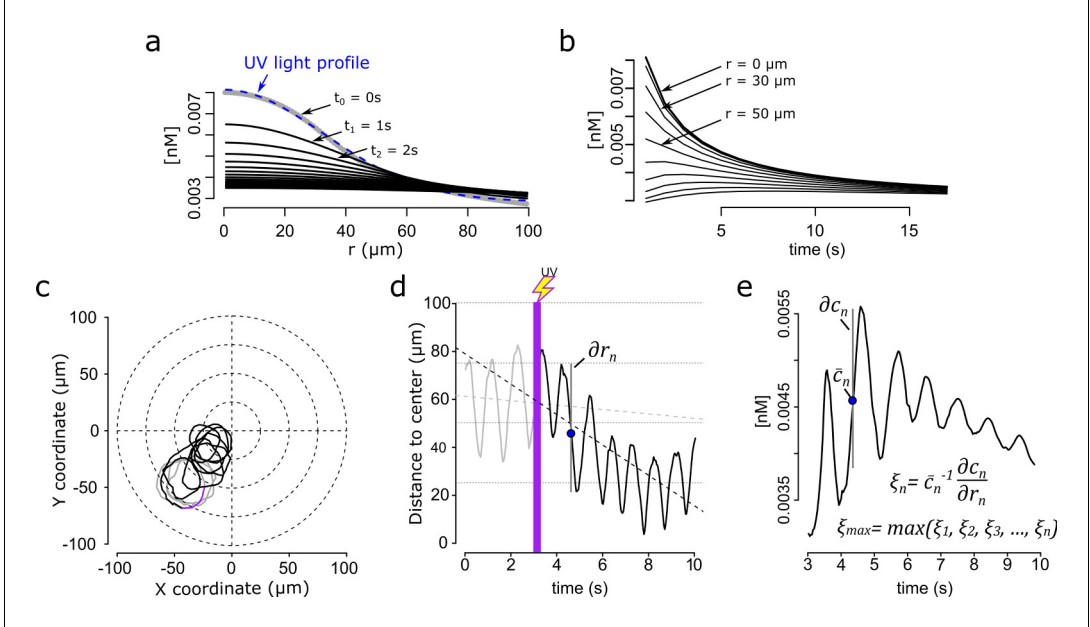

**Figure 5.** Steep speract gradients provoke chemotaxis in *S. purpuratus* spermatozoa. a. Dynamics of the *f2* speract gradient. The blue dashed line ($t_0 = 0$ s) corresponds to a Gaussian distribution fitted to the UV light profile and illustrates the putative shape of the instantaneously-generated speract concentration gradient. Solid black lines illustrate the temporal evolution of the speract concentration field after t = 1, 2, 3, . . ., 20 s. (b) Temporal changes in the *f2* speract field computed radially (each 10 μm) from the center of the gradient. (c) Characteristic motility changes of a *S. purpuratus* spermatozoon exposed to the *f2* speract gradient. Solid lines illustrate its swimming trajectory 3 s before (gray), during UV flash (purple) and 6 s after (black) speract exposure. (d) Spermatozoa head distance to the source of the speract gradient versus time, calculated from sperm trajectory in (c). (e). Stimulus function computed from the swimming behavior of the spermatozoon in (c), considering the dynamics of (a and b).

The online version of this article includes the following figure supplement(s) for figure 5:

**Figure supplement 1.** Modeling of the dynamics of speract gradient based on the UV light profile of distinct optical fibers.

**Figure supplement 2.** Characteristic motility changes of a *S. purpuratus* spermatozoon exposed to *f3* and *f4* speract gradients (chemotactic *vs.* non-chemotactic response).

*Berg and Purcell (1977)*, which considers the minimal requirements needed for a single searcher (i.e. a sperm cell) to gather sufficient information to determine the orientation of a non-uniform concentration field. By considering the difference between *L. pictus* and *S. purpuratus* spermatozoa in terms of the number of chemoattractant receptors, receptor pocket effective size, cell size, sampling time, mean linear velocity, sampling distance, and the local mean and slope of the chemoattractant concentration gradient, our model predicts that *S. purpuratus* spermatozoa would need a speract gradient three times steeper than the gradient that drives chemotactic responses for *L. pictus* spermatozoa. We tested this experimentally by exposing *S. purpuratus* spermatozoa to various defined speract concentration gradients.

We showed that *S. purpuratus* spermatozoa can undergo chemotaxis, but only if the speract concentration gradients are sufficiently steep, as predicted by the chemoreception model (i.e. speract gradients that are at in the region of three times steeper than the speract concentration gradient that drives chemotaxis in *L. pictus* spermatozoa). This confirms and explains why the shallower speract gradients previously tested are unable to generate any chemotactic response in *S. purpuratus* spermatozoa (*Guerrero et al., 2010a*), despite inducing characteristic 'turn and run' motility responses.

These findings indicate that the guiding chemical gradient must have a minimum steepness to elicit sperm chemotaxis, where the signal-to-noise relationship (*SNR*) of stimulus to the gradient detection mechanism imposes a limit for the chemotactic efficiency. Our results are in agreement with recent theoretical studies by Kromer and colleagues, indicating that sperm chemotaxis of marine invertebrates operates optimally within a boundary defined by the *SNR* of collecting ligands within a chemoattractant concentration gradient (*Kromer et al., 2018*). We showed that *SNR* can be

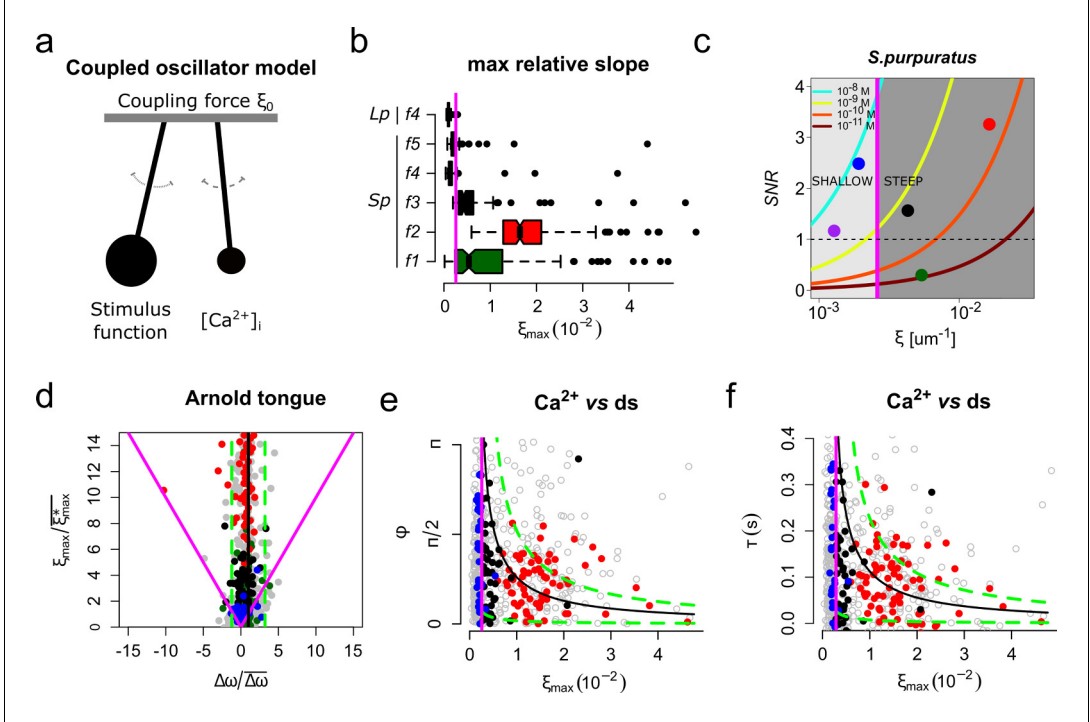

**Figure 6.** The slope of the speract concentration gradient generates a frequency-locking phenomenon between the stimulus function and the internal Ca²⁺ oscillator triggered by speract. (a) Coupled oscillator model. Each sperm has two independent oscillators: i) stimulus function and ii) $[Ca^{2+}]_i$, which can be coupled through a forcing term that connects them, in our case the slope of the chemoattractant concentration gradient ($\xi_0$). (b) Maximum relative slopes ($\xi_{max}$) of the chemoattractant concentration gradient experienced by *S. purpuratus* (*Sp*) spermatozoa when exposed to *f1, f2, f3, f4, f5* speract gradients. The maximum relative slopes of the chemoattractant concentration gradient experienced by *L. pictus* spermatozoa (*Lp*) toward *f4* experimental regime are also shown. Note that $\xi_{max}$ for *f2, f3,* and *f5*, are up to 2–3 times greater than in *f4*, regardless of the species. (c) Experimental signal-to-noise ratios (*SNR*) regimes experienced by spermatozoa swimming in different gradient conditions. Note that only *f2, f3* and *f5* have higher *SNR*, compared to other gradient conditions, for which stochastic fluctuations mask the signal. This *SNR* calculation assumes a 10% of speract uncaging. The maximum relative slopes ($\xi$) are shown in log scale (d) Arnold's tongue indicating the difference in intrinsic frequency of the internal Ca²⁺ oscillator of *S. purpuratus* spermatozoa, just before and after the speract gradient exposure. (e). Phase difference between the time derivative of the stimulus function and the internal Ca²⁺ oscillator of *S. purpuratus* spermatozoa, obtained by computing the cross-correlation function between both time series (*Figure 6—figure supplement 2*). (f). Phase difference between the time derivative of the stimulus function and the internal Ca²⁺ oscillator of *S. purpuratus* spermatozoa expressed in temporal delays. (d-f) Gray points represent the collated data of all *f1, f2, f3, f4, f5* experimental regimes. Red, black and blue points indicate chemotactic spermatozoa (CI > 0 at second three after UV flash), located in R3, and R4 regions just before the speract gradient is established under *f2, f3* and *f5* experimental regimes, respectively. Magenta lines represent the transition boundary ($\gamma_{min} = \overline{\xi^*_{max}} \sim 2.6 \times 10^{-3}$ μm⁻¹, see also *Figure 1d–f*) below which no synchrony is observed, obtained from the theoretical estimates (black curves, mean of $\Delta\omega$) of panels (e) and (f). Green dashed lines indicate confidence intervals (mean ± standard deviation).

The online version of this article includes the following figure supplement(s) for figure 6:

**Figure supplement 1.** Speract induces Ca²⁺ oscillations in immobilized *S. purpuratus* spermatozoa.

**Figure supplement 2.** Cross-correlation analysis of $[Ca^{2+}]_i$ and stimulus function derivative (dS) signals.

tuned by the steepness of the chemical gradient, where higher *SNR*'s are reached at steeper gradients, hence increasing the probabilities of locating the source of the gradient.

The large majority of marine spermatozoa characterized to date, together with many motile microorganisms, explore their environment via helical swimming paths, whereupon encountering a surface these helices collapse to circular trajectories. The intrinsic periodicity of either swimming behavior commonly results in the periodic sampling of the cell chemical environment with direct implications for their ability to accurately perform chemotaxis.

The periodic sampling of chemoattractants by the sperm flagellum continuously feeds back to the signaling pathway governing the intracellular Ca²⁺ oscillator, hence providing a potential coupling mechanism for sperm chemotaxis. Indirect evidence for the existence of a feedback loop operating between the stimulus function and the Ca²⁺ oscillator triggered by chemoattractants has been found

in *L. pictus*, *A. punctulata* and *Ciona intestinalis* (ascidian) species, whose spermatozoa show robust chemotactic responses toward their conspecific chemoattractants (*Böhmer et al., 2005*; *Guerrero et al., 2010a*; *Jikeli et al., 2015*; *Shiba et al., 2008*).

To investigate further the molecular mechanism involved in sperm chemotaxis, we measured both the stimulus function and the triggered $[Ca^{2+}]_i$ oscillations for up to one thousand *S. purpuratus* spermatozoa exposed to five distinctly-shaped speract concentration gradients. We demonstrate that the steepness of the slope of the chemoattractant concentration gradient is a major determinant for sperm chemotaxis in *S. purpuratus* and might be an uncovered feature of sperm chemotaxis in general. A steep slope of the speract concentration gradient entrains the frequencies of the stimulus function and the internal $Ca^{2+}$ oscillator triggered by the periodic sampling of a non-uniform speract concentration field. We assessed the transition boundary of the coupling term (the slope of the speract concentration gradient) for the two oscillators to synchronize and found it to be very close to the boundary where *S. purpuratus* starts to experience chemotaxis. The agreement of our data with a model of weakly-coupled phase oscillators suggests that the slope of the speract concentration gradient is the driving force behind the observed synchronous oscillations and, as a result, for the chemotactic ability of sea urchin spermatozoa.

It is not that surprising to find matching of frequencies when dealing with two oscillators coupled through a forcing term. Nonetheless, the boundaries of the 'region of synchrony' are by no means trivial. What is relevant to the former discussion is the existence of thresholds in the coupling strength, whose experimental calculations agree with our theoretical predictions based on the chemoreception model. In addition, such a minimal model for coupled oscillators is also able to predict computed functional dependencies that are well documented in the literature, that is the observed temporal and frequency lags between the stimulation and signaling responses of the chemoattractant signaling pathway (*Alvarez et al., 2012*; *Böhmer et al., 2005*; *Guerrero et al., 2010a*; *Kaupp et al., 2003*; *Nishigaki et al., 2004*; *Pichlo et al., 2014*; *Shiba et al., 2008*; *Strünker et al., 2006*; *Wood et al., 2007*; *Wood et al., 2005*).

Caution must be exercised with the interpretations of the agreement of our data with such a generic model for coupled phase oscillators, particularly when considering only a few steps of the oscillatory cycles. The latter is relevant for assessing frequency entrainment, which in some cases demands a certain delay before reaching the synchronized state, that is when the natural frequencies of the connected oscillators are very distinct. The chemotactic responses scored in the present study encompass a few steps (<10) of both the stimulus function and the internal $Ca^{2+}$ oscillator triggered by speract (*Figure 5—figure supplement 2*, *Figure 6—figure supplement 1* and *Figure 6—figure supplement 2*). Our data indicate that within the chemotactic regime, frequency entrainment of the stimulus function and the internal $Ca^{2+}$ oscillator of *S. purpuratus* spermatozoa seems to occur almost instantaneously, within the first three oscillatory steps (*Figure 6—figure supplement 2*). Such interesting findings can be explained by the proximity of the natural frequencies of both oscillators (*Figure 6d*), which may relieve the need for a longer delay for reaching frequency entrainment. Whether the proximity of the frequencies of both oscillators is sculped by the ecological niche where sperm chemotaxis occurs is an open question, however, a near-instantaneous entrainment would confer obvious evolutionary advantage under the reproductively competitive conditions of synchronized spawning as undertaken by sea urchins.

One can further hypothesize about the evolutionary origin of the described differences in sensitivity to chemoattractant concentration gradients between *S. purpuratus* and *L. pictus* spermatozoa if we consider their respective ecological reproductive niches. The turbulent environment where sea urchins reproduce directly impinges on the dispersion rates of small molecules such as speract, hence, imposing ecological limits that constrain permissive chemoattractant gradient topologies within different hydrodynamic regimes. For instance, the reproductive success of *L. pictus*, *S. purpuratus* and abalone species has been shown to peak at defined hydrodynamic shearing values (*Hussain et al., 2017*; *Mead and Denny, 1995*; *Riffell and Zimmer, 2007*; *Zimmer and Riffell, 2011*). What are the typical values of the chemoattractant gradients encountered by the different species in their natural habitat? The correct scale to consider when discussing the small-scale distribution of chemicals in the ocean is the Batchelor scale, $l_B = (\eta D^2/\zeta)^{1/4}$, where $\eta$ is kinematic viscosity, $D$ the diffusion coefficient and $\zeta$ is the turbulent dissipation rate (*Aref et al., 2017*; *Batchelor et al., 1959*). Turbulence stirs dissolved chemicals in the ocean, stretching and folding them into sheets

and filaments at spatial dimensions down to the Batchelor scale: below $l_B$ molecular diffusion dominates and chemical gradients are smoothened out.

*S. purpuratus* is primarily found in the low intertidal zone. The purple sea urchin lives in a habitat with strong wave action and areas with shaking aerated water. These more energetic zones, including tidal channels and breaking waves, generate relatively high levels of turbulence ($\zeta \sim 10^{-4}$ m$^2$s$^{-3}$) which lead to somewhat small values of $l_B$ and, hence, to steep gradients (i.e. $1/l_B$). *L. pictus*, on the contrary, is mostly found at the edge of or inside kelp beds, well below the low tide mark where the levels of turbulence are much more moderate ($\zeta \sim 10^{-6}$ m$^2$s$^{-3}$) (*Jimenez, 1997*; *Thorpe, 2007*). This difference in the turbulent kinetic energy dissipation rate has a significant effect on the spatial dimensions of chemical gradients for sperm chemotaxis present in a particular habitat. The ratio of $l_B$ for the different habitats scales as $l_{Bpurpuratus}/l_{Bpictus} \sim (\zeta_{pictus}/\zeta_{purpuratus})^{1/4} \sim 3$, which fits considerably well with the relative sensitivity to speract of the two species. Furthermore, we have shown that *S. purpuratus* spermatozoa experience chemotaxis toward steeper speract gradients than those that guide *L. pictus* spermatozoa, which is also compatible with the distinct chemoattractant gradients they might naturally encounter during their journey in search of an egg.

## Materials and methods

### Materials
Artificial seawater (ASW), and Low Ca$^{2+}$ ASW were prepared as in *Guerrero et al. (2010a)*, their detailed composition, together with an extended list of other materials is presented in the Appendix 1. Caged speract (CS), was prepared as described previously (*Tatsu et al., 2002*).

### Loading of Ca$^{2+}$-fluorescent indicator into spermatozoa and microscopy imaging
*S. purpuratus* or *L. pictus* spermatozoa were labeled with fluo-4-AM (as described in section 2.2. Loading of Ca$^{2+}$-fluorescent indicator into spermatozoa in Appendix 1), and their swimming behavior was studied at the water-glass interface on an epifluorescence microscope stage (Eclipse TE-300; Nikon). The cover slips were covered with poly-HEME to prevent the attachment of the cells to the glass. Images were collected with a Nikon Plan Fluor 40×/1.3NA oil-immersion objective. Temperature was controlled directly on the imaging chamber at a constant 15°C. Stroboscopic fluorescence excitation was provided by a Cyan LED synchronized to the exposure output signal of the iXon camera (2 ms illumination per individual exposure, observation field of 200×200 μm), the fluorescence cube was set up accordingly (see Appendix 1).

### Image processing and quantification of global changes of spermatozoa number and [Ca$^{2+}$]$_i$
To study the dynamics of overall sperm motility and [Ca$^{2+}$]$_i$ signals triggered by the distinct speract gradients, we developed an algorithm that provides an efficient approach to automatically detect the head of every spermatozoa in every frame of a given video-microscopy file. A detailed description of the algorithm is provided in the Appendix 1.

### Computing the dynamics of speract concentration gradients
The dynamics of the chemoattractant gradient was computed using Green's function of the diffusion equation, considering diffusion in 2D:

$$c = f(r,t) = \frac{C_0}{4\pi D(t+t_0)} e^{\frac{-r^2}{\sigma^2}} + c_b, \tag{5}$$

*Equation (5)* for the concentration tells us that the profile has a Gaussian form, where $D$ is the diffusion coefficient of the chemoattractant, $c_b$ is the basal concentration of the chemoattractant, $t$ is the time interval, $r$ is the distance to the center of the gradient and $c_0$ is the initial concentration. The width of the Gaussian is $\sigma = \sqrt{4D(t+t_0)}$, and hence it increases as the square root of time.

The speract concentration gradients were generated via the photolysis of 10 nM caged speract (CS) with a 200 ms UV pulse delivered through each of four different optical fibers with internal

**Table 1.** Sperm accumulation responses triggered by different speract gradients.

| Optical fiber | Sperm accumulation at the central regions of the imaging field | Sperm depleted of distal regions of the imaging field | $[Ca^{2+}]_i$ rise (fold) |
|---|---|---|---|
| f1 | No | No | <2 |
| f2 | R1 and R2 | R3 and R4 | >2 |
| f3 | R1, R2 and R3 | R4 | >2 |
| f4 | No | No | >2 |
| f5 | No | No | ~2 |

Accumulation responses were evaluated at second 6, that is 3 s after photo-liberation of speract by a 200 ms UV flash.

diameters of 0.2, 0.6, 2, and 4 mm (at two different positions). Light intensity was normalized dividing each point by the sum of all points of light intensity for each fiber and multiplying it by the fiber potency (measured at the back focal plane of the objective) in milliwatts (mW) (*Supplementary file 2*). Each spatial distribution of instantaneously-generated speract concentration gradient was computed by fitting their corresponding normalized spatial distribution of UV light (Residual standard error: $2.7 \times 10^{-5}$ on 97 degrees of freedom), considering an uncaging efficiency of 5–10%, as reported (*Tatsu et al., 2002*).

The diffusion coefficient of speract has not been measured experimentally. However, the diffusion coefficient of a similar chemoattractant molecule, resact (with fourteen amino acids), has been reported, $D_{resact} = 239 \pm 7 \ \mu m^2 \ s^{-1}$ (*Kashikar et al., 2012*). If we consider that speract is a decapeptide, the 1.4 fold difference in molecular weight between speract and resact would imply a $(1.4)^{1/3}$ fold difference in their diffusion coefficients, which is close to the experimental error reported (*Kashikar et al., 2012*). For the sake of simplicity, the spatio-temporal dynamics of the distinct instantaneously generated speract gradients was modeled considering a speract diffusion coefficient of $D_{speract} = 240 \ \mu m^2 \ s^{-1}$.

## Computing $[Ca^{2+}]_i$ dynamics and the stimulus function of single spermatozoa

Spermatozoa were tracked semi-automatically by following the head centroid with the MtrackJ plugin (*Meijering et al., 2012*) of ImageJ 1.49u. Single cell $[Ca^{2+}]_i$ signals were computed from the mean value of a 5x5 pixel region, centered at each sperm head along the time. The head position of each spermatozoa $x$ was used to compute the mean concentration of speract at $r$ over each frame. The stimulus function of single spermatozoa $S = f(c)$ was computed by solving *Equation (5)* considering both their swimming trajectories, and the spatio-temporal evolution of a given speract concentration gradient. The profiles of UV light were used to compute the initial conditions at $c(r, t_o)$.

The phase- and temporal-shifts between the time derivative of the stimulus function $dS/dt$ and the internal $Ca^{2+}$ oscillator triggered by speract, were computed from their normalized cross-correlation function.

Programs were written in R statistical software (*R Development Core Team, 2016*).

## Chemotactic index (CI)

Each sperm trajectory was smoothened using a moving average filter, with a window of 60 frames (two seconds approximately) (*Figure 4b* and *Video 2*). A linear model was then fitted to the smoothed trajectory; the corresponding line was forced to go through the mean point of the smoothed trajectory (orange point in *Figure 4b* and *Video 2*). The θ angle between red and black vectors was calculated in each frame from the second 4.5 to 10.

The chemotactic index is defined based on the progressive displacement of the sperm trajectory as $CI = \frac{|u|cos\theta - |v|cos\varphi}{|u| + |v|}$, being $\phi$ and $\theta$ the angles between gray and magenta, and red and black vectors, respectively; and $|v|$ and $|u|$ the magnitude of the sperm progressive speed before and after speract uncaging, respectively (*Figure 4b* and *Video 2*). The CI considers the sperm displacement before speract uncaging (i.e. unstimulated drift movement at 0–3 s), and then subtracts it from the speract induced effect (at 3–10 s). The CI takes values from −1 (negative chemotaxis) to 1 (positive chemotaxis), being 0 no chemotaxis at all.

## Statistical analyses

The normality of the CI distributions, each obtained from *f1* to *f5* speract gradient stimuli, was first assessed using the Shapiro-Wilk test; none of them were normal (Gaussian), so each CI distribution was analyzed using non-parametric statistics (*Figure 4d* and *Appendix 1—video 7*). The curves obtained from medians of each CI distribution were smoothed using a moving average filter, with a window of 20 frames (0.6 s) (*Figure 4c*).

Data are presented for individual spermatozoa (n) collected from up to three sea urchins. All statistical tests were performed using R software (*R Development Core Team, 2016*). The significance level was set at 95% or 99%.

## Acknowledgements

The authors thank Dr. Tatsu Yoshiro for providing the caged speract, and Drs. Hermes Gadêlha, David Smith and Nina Pastor for stimulating discussions and a critical reading of the manuscript. AG thanks Dr. Manabu Yoshida and Dr. Kaoru Yoshida for feedback regarding sperm chemotaxis in marine invertebrates, and to the Japan Society for the Promotion of Science (JSPS invitation fellowship for research in Japan to A.G., short term JSPS/236, ID no. S16172). AD performed part of this work while carrying out a Sabbatical at the Instituto Gulbenkian de Ciencia (IGC) supported by UNAM/DGAPA and IGC.

## Additional information

### Funding

| Funder | Grant reference number | Author |
| --- | --- | --- |
| Consejo Nacional de Ciencia y Tecnología | Fronteras 71 | Alberto Darszon<br>Adán Guerrero |
| Consejo Nacional de Ciencia y Tecnología | Ciencia basica 252213 y 255914 | Alberto Darszon<br>Adán Guerrero |
| Dirección General de Asuntos del Personal Académico, Universidad Nacional Autónoma de México | IA202417 | Adán Guerrero |
| Dirección General de Asuntos del Personal Académico, Universidad Nacional Autónoma de México | IN205516 | Alberto Darszon |
| Dirección General de Asuntos del Personal Académico, Universidad Nacional Autónoma de México | IN206016 | Carmen Beltran |
| Dirección General de Asuntos del Personal Académico, Universidad Nacional Autónoma de México | IN215519 | Carmen Beltran |
| Dirección General de Asuntos del Personal Académico, Universidad Nacional Autónoma de México | IN112514 | Alberto Darszon |
| Ministerio de Economía y Competitividad | FIS2013-48444-C2-1-P | Idan Tuval |
| Ministerio de Economía y Competitividad | FIS2016-77692-C2-1- P | Idan Tuval |
| Japan Society for the Promotion of Science | JSPS/236, ID no. S16172 | Adán Guerrero |

The funders had no role in study design, data collection and interpretation, or the decision to submit the work for publication.

## Author contributions
Héctor Vicente Ramírez-Gómez, Conceptualization, Data curation, Software, Formal analysis, Validation, Investigation, Visualization, Methodology, Writing - review and editing, developed the chemotactic index; Vilma Jimenez Sabinina, Formal analysis, Investigation, Visualization, Methodology, performed the experiments; Martín Velázquez Pérez, Methodology; Carmen Beltran, Supervision, Funding acquisition; Jorge Carneiro, Conceptualization, Supervision, Validation; Christopher D Wood, Supervision, Writing - review and editing; Idan Tuval, Conceptualization, Formal analysis, Supervision, Funding acquisition, Validation, Methodology, Writing - review and editing; Alberto Darszon, Conceptualization, Resources, Supervision, Funding acquisition, Writing - original draft, Project administration, Writing - review and editing; Adán Guerrero, Conceptualization, Software, Formal analysis, Supervision, Funding acquisition, Validation, Investigation, Methodology, Writing - original draft, Project administration, Writing - review and editing, performed the experiments

## Author ORCIDs
Héctor Vicente Ramírez-Gómez ⓘ https://orcid.org/0000-0002-4526-4689
Vilma Jimenez Sabinina ⓘ https://orcid.org/0000-0002-9390-1646
Carmen Beltran ⓘ http://orcid.org/0000-0001-9344-7618
Jorge Carneiro ⓘ https://orcid.org/0000-0002-7520-3406
Idan Tuval ⓘ https://orcid.org/0000-0002-6629-0851
Adán Guerrero ⓘ https://orcid.org/0000-0002-4389-5516

## Ethics
Animal experimentation: All of the animals were handled according to approved institutional animal care and use committee protocols (# 44, 142, 188, 193, 285) of the Instituto de Biotecnología of the Universidad Nacional Autónoma de México.

## Decision letter and Author response
Decision letter https://doi.org/10.7554/eLife.50532.sa1
Author response https://doi.org/10.7554/eLife.50532.sa2

# Additional files

### Supplementary files
• Supplementary file 1. Parameters of the chemoattractant sampling model for each species. Note that the main differences between species are the number of receptors $N$. $N_{1/2}$ number of receptors that allows half maximal binding rate for any concentration of chemoattractant, that is $\pi a/s$. $D$ diffusion coefficient of the chemoattractant; $Kon$ association rate constant; $s$ effective radius of the chemoattractant (as proxy of chemoattractant receptor's binding site radius); $\Delta t$ sampling time (time to swim half the circumference in the boundary close to the water-glass interface); $v$ mean linear speed of the spermatozoa, $\Delta r/\Delta t$; $\Delta r$ sampling distance (circumference diameter); $L$ length of sperm flagellum; $a$ spermatozoa radius, assuming that flagella are spheres; $Pe$ Peclet number for a spherical cell approximation (sphere), or cylindrical flagellum geometry (cylinder). [a] Measured in this study (mean ± sd); N = 3 sea urchins; n = 495 (*S. purpuratus*), n = 56 (*L. pictus*) spermatozoa. [b]*Nishigaki and Darszon (2000)*; *Nishigaki et al. (2001)*. [c] Calculated in this study (see section 1.1. On the estimate of maximal chemoattractant absorption). [d] Measured in this study (mean ± sd); N = 1 sea urchin; n = 26 (*S. purpuratus*), n = 39 (*L. pictus*) spermatozoa. [e]*Kashikar et al. (2012)*. [f]*Pichlo et al. (2014)* reported 6.5 x $10^{-8}$ cm for the resact radius. [g] Calculated in this study.

• Supplementary file 2. Physical diameter of the optical fibers, and UV light power measured at the back focal plane of the objective. *Typically, there is an extra 20% loss of light power between the back focal plane of the objective and the sample, due to scattering within the optics.

• Transparent reporting form

## Data availability

All data generated or analyzed during this study are included in the manuscript and supporting files.

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

## Appendix 1

### 1.Theory

#### 1.1. On the estimate of maximal chemoattractant absorption

*Berg and Purcell (1977)* derived a simple expression for the mean chemoattractant binding and absorption flux by a cell in the steady-state, denoted $J$:

$$J = 4\pi Da\bar{c}\frac{N}{N\ +\ \pi a/s} = J_{max}\frac{N}{N\ +\ \pi a/s} \qquad (A1)$$

where $D$ is the diffusion coefficient of the chemoattractant; $a$ is the radius of the cell; $\bar{c}$ is the mean concentration of the chemoattractant; $N$ is the number of receptors on the membrane of the cell; $s$ is the effective radius of the receptor, assumed to be disk-like on the cell surface and binding to chemoattractant molecules with high affinity; $J_{max} = 4\pi Da\bar{c}$, which is the maximal flux of chemoattractant that a cell in the steady-state can experience; and the receptor term $\frac{N}{N+\pi a/s}$ is the probability that a molecule that has collided with the cell will find a receptor (*Berg and Purcell, 1977*).

The receptor term arises from the matching of two distinct limits: for a low number of receptors, the flux into independent patches leads to an overall diffusive flux into the sphere that is linear with the number of receptors. In the opposite limit of large surface coverage, the 'interactions between the effects of adjacent receptors' leads indeed to the saturation of chemoreception. The expression implies that for $N\ \gg \pi a/s =\ N_{1/2}$ the flux of chemoattractant absorption becomes $J\ \cong J_{max}$, which means that coverage of only a small fraction of the cell surface by the receptors may lead to maximal flux. The flux becomes practically independent of the number of receptors, proportional to the concentration of the chemoattractant and limited by its diffusion, when receptor density is sufficiently large. For a given chemoattractant concentration the half maximal flux $\left(J = \frac{1}{2}J_{max}\right)$ is reached when the number of receptors is $N =\ N_{1/2} =\ \pi a/s$.

It is worth getting a rough estimate of the number of receptors required for a maximal influx of chemoattractant, in the specific case of spermatozoa by calculating $N_{1/2}$, assuming a spherical cell with a surface area equivalent to that of the actual flagella. In the case of *S. purpuratus* sperm, the flagellar width h $\cong$ 0.2 μm, and length L $\cong$ 40 μm that would give us an approximate surface area A ~ 25 μm$^2$ or an equivalent spherical radius $a_e$ ~ 1.4 μm and, hence, a $J_{max} \cong 0.1 J_{max}^{sph}$, where $J_{max}^{sph}$ represents the maximal influx of chemoattractants for the spherical cell. The factor 0.1 relating the maximal flux $J_{max}$ in a cylindrical flagellum and $J_{max}^{sph}$ in a spherical cell, arises by recalling that the expression for $J_{max}$ in *Equation (A1)*.

The later stems from an analogy with electrostatics such that the total current depends on the electrical capacitance C of the conducting material and, in particular, on the geometrical arrangement. The capacitance of a simple spherical conductor equals the radius $a$ of the sphere but more generally we have $J_{max} = 4\pi CD\bar{c}$ (*Berg and Purcell, 1977*).

Note that the spherical geometry is a first order approximation, which has been extremely useful and successful in the past in shedding light on many problems with more complex geometries. This includes the first estimate of diffusive fluxes in this same chemotaxis problem (as Berg and Purcell showed in 1977). Here, we have followed the same principle of 'minimal modelling' that captures the main physics but that, at the same time, allows for simple characterization of the relevant parameters (e.g. the dependence with the number of receptors).

A more accurate computation can be obtained by considering the nearly cylindrical shape of the flagellum. The capacitance of a finite cylinder can be obtained as a series expansion in the logarithm of the cylinder aspect ratio $\Lambda$ = ln(L/h) (*Maxwell, 1877*) and, to a second order in $1/\Lambda$, and is given by the following expression:

$$C = \frac{2\pi L}{\Lambda}\left[1 + \frac{1}{\Lambda}(1 - ln2) + \frac{1}{\Lambda^2}\left\{1 + (1 - ln2)^2 - \frac{\pi^2}{12}\right\} + O\left(\frac{1}{\Lambda^3}\right)\right] \quad \text{(A2)}$$

For the case of the slender sperm flagellum with h/L << 1, $\Lambda \cong 5$, this expression gives a maximal influx for the cylinder ($J_{max}^{cyl}$) that is, again, approximately one tenth that of the equivalent sphere:

$$J_{max}^{cyl} \cong \frac{2\pi}{5}LD\bar{c} \cong 0.1 J_{max}^{sph} \quad \text{(A3)}$$

The above description is valid only in the limit of instantaneous adsorption at the receptors. For a finite rate of binding by the receptors we can simply modify the above expressions to include an effective size for the binding sites $se = kon/D$ (**Phillips et al., 2012**). With $kon = 24$ $\mu M^{-1}s^{-1}$ and $27$ $\mu M^{-1}s^{-1}$ being the corresponding affinity constants for speract and its receptor, calculated by Nishigaki in 2000 for *L. pictus* and in 2001 for *S. purpuratus* (**Nishigaki and Darszon, 2000**; **Nishigaki et al., 2001**), respectively. $D = 240$ $\mu m^2 s^{-1}$, $se = 1.7$ Å and $1.9$ Å, respectively, which is indeed much smaller than the physical size of the receptors (**Supplementary file 1**). Note that the dimensions of the speract receptor radius are not known, however **Pichlo et al. (2014)** provided an estimation of the radius of the resact receptor (the extracellular domain of the GC) of 2.65 nm. The value of s ~ 0.19 nm used in this work is about one order of magnitude smaller than such estimation. This value arises not from estimates of either receptor or chemoattractant sizes, but rather from an estimate of the effective size of the binding site, based on experimental measurements of chemoattractant binding kinetics.

These equivalences were used to obtain the estimates in **Supplementary file 1**, which are discussed in the main text. From these estimates, we can compute $N_{1/2} \sim 3\times10^4$ as the total number of SAP receptors for the *S. purpuratus* sperm flagellum to act as a perfect absorber. As the actual number of SAP receptors for this species is lower than that figure, that is $N < N_{1/2}$, we cannot approximate the solution of **Equation (A1)** to that of a perfect absorber. More specifically, under these circumstances the absorption remains almost linearly dependent on the actual number of receptors on the flagellum (**Figure 1—figure supplement 1**).

## 1.2. A condition for detecting a change in the chemoattractant concentration

A cell uses the chemoattractant it samples from the medium as a proxy of the extracellular concentration of the chemoattractant at any given time. The flux of chemoattractant $J$, calculated in the previous section, measures the sampling rate. Because the number of chemoattractant molecules is finite and small, the actual number of molecules sampled by the cell in an interval of time $\Delta t$ is a random variable, denoted $n$, which is Poisson distributed with expected value $E[n] = J\Delta t$ and standard deviation $SD[n] = \sqrt{J\Delta t}$. The chemotaxis signaling system of the spermatozoon should remain unresponsive while the cell is swimming in an isotropic chemoattractant concentration field, as there are no spatial cues for guidance (**Figure 1a**), although motility responses may still be triggered. For example, the stereotypical turn-and-run motility responses of *S. purpuratus* sperm in the presence of speract (isotropic fields or weak gradients) (**Wood et al., 2007**)., it has been previously reported that the turn-and-run motility response is necessary, but not sufficient, for sea urchin sperm chemotaxis (**Guerrero et al., 2010a**). The Poisson fluctuations of sampled chemoattractant molecules, measured by $\sqrt{J\Delta t}$ can be understood as background noise and hence should not elicit a response. When the sperm is swimming confined to a plane in a chemoattractant gradient produced by the egg (**Figure 1ab and c**), the chemotactic responses should be triggered only when the amplitude of the sampling fluctuations are sufficiently large as to not be confused with the background noise, that is when the difference in concentration at the two extremes of the circular trajectory leads to fluctuations in chemoattractant sampling that are larger than the background noise. These considerations lead to a minimal condition for reliable detection

of a chemotactic signal (**Berg and Purcell, 1977**; **Vergassola et al., 2007**), the corresponding condition can be stated as:

$$\left(4\pi Da\bar{c}\,\frac{N}{N+\pi a/s}\Delta t\right)v\Delta t\frac{\partial c}{\partial r}\bar{c}^{-1} > \sqrt{4\pi Da\bar{c}\,\frac{N}{N+\pi a/s}\Delta t} \tag{A4}$$

where $E[n]=J\Delta t=4\pi Da\bar{c}\,\frac{N}{N+\pi a/s}\Delta t$; $\Delta t$ is specifically the time the sperm takes to make half a revolution in its circular trajectory; $v$ is the mean linear velocity, defined as $v=\frac{\Delta r}{\Delta t}$, where $\Delta r$ is the diameter of the circumference in the 2D sperm swimming circle (**Figure 1c** and **Supplemnetary file 1**); and $\xi=\bar{c}^{-1}\frac{\partial c}{\partial r}$ is the relative slope of the chemoattractant concentration gradient.

As described in the main text, by interpreting the left-hand side of the **Equation (A4)** as the minimal chemotactic signal; and the right-hand side as a measurement of the background noise at a given mean concentration. Hence, one can obtain a minimal condition for the smallest signal to noise ratio (SNR) necessary to elicit a chemotactic response. **Equation (A4)** can be rewritten in terms of signal-to-noise ratio:

$$SNR = v\Delta t^{3/2}\left(4\pi Da\bar{c}\,\frac{N}{N+\pi a/s}\right)^{1/2}\xi > 1 \tag{A5}$$

Note that all previous **Equations (A1-A5)** are only valid for small Peclet numbers ($Pe \leq 1$) which is indeed the case for chemoattractant transport to the sperm. Pe estimates the relative importance of advection (directed motion) and diffusion (random-like spreading) of 'anything that moves'. We are studying the motion of chemoattractant molecules: they are transported (relative to the swimming sperm) by its swimming while jiggling around by Brownian motion at the molecular scale.

An evidence-based estimate of the Peclet number for chemoattractants can be provided by following the definition of the Peclet number Pe = UR/D, with the sperm swimming speed in the range U ~ [72–100 µm s$^{-1}$], diffusivity D ~ 240 µm$^2$ s$^{-1}$ for the chemoattractant. The critical length scale R for the diffusive problem can be estimated by either i) computing the influx transport problem in a cylindrical geometry with the fluid flow parallel to the flagellar long axis (i.e. the sperm swimming direction) for which R is the flagellar width ~0.2 µm; or ii) for the simplified spherical cell approximation for which R is simply the equivalent spherical radius $a_e$ ~ [1.39–1.58 µm] (see section **1.1. On the estimate of maximal chemoattractant absorption**). This renders Pe ~ [6e$^{-2}$ - 6e$^{-1}$] $\leq$ 1 for all experiments presented in this manuscript.

# 2.Extended materials and methods

## 2.1. Materials

Undiluted *S. purpuratus* or *L. pictus* spermatozoa (JAVIER GARCIA PAMANES, Ensenada, Mexico PPF/DGOPA224/18 Foil 2019, RNPyA 7400009200; and South Coast Bio-Marine San Pedro, CA 90731, USA respectively) were obtained by intracoelomic injection of 0.5 M KCl and stored on ice until used within a day. Artificial seawater (ASW) was 950 to 1050 mOsm and contained (in mM): 486 NaCl, 10 KCl, 10 CaCl$_2$, 26 MgCl$_2$, 30 MgSO$_4$, 2.5 NaHCO$_3$, 10 HEPES and 1 EDTA (pH 7.8). For experiments with *L. pictus* spermatozoa, slightly acidified ASW (pH 7.4) was used to reduce the number of spermatozoa experiencing spontaneous acrosome reaction. Low Ca$^{2+}$ ASW was ASW at pH 7.0 and with 1 mM CaCl$_2$, and Ca$^{2+}$-free ASW was ASW with no added CaCl$_2$. [Ser5; nitrobenzyl-Gly6]speract, referred to throughout the text as caged speract (CS), was prepared as previously described (**Tatsu et al., 2002**). Fluo-4-AM and pluronic F-127 were from Molecular Probes, Inc (Eugene, OR, USA). PolyHEME [poly(2-hydroxyethylmethacrylate)] was from Sigma-Aldrich (Toluca, Edo de Mexico, Mexico).

## 2.2. Loading of Ca²⁺-fluorescent indicator into spermatozoa

This was done as in *Beltrán et al. (2014)*, as follows: undiluted spermatozoa were suspended in 10 volumes of low $Ca^{2+}$ ASW containing 0.2% pluronic F-127 plus 20 µM of fluo-4-AM and incubated for 2.5 hr at 14°C. Spermatozoa were stored in the dark and on ice until use.

## 2.3. Imaging of fluorescent swimming spermatozoa

The cover slips were briefly immersed into a 0.1% wt/vol solution of poly-HEME in ethanol, hot-air blow-dried to rapidly evaporate the solvent, wash with distilled water twice followed by ASW and mounted on reusable chambers fitting a TC-202 Bipolar temperature controller (Medical Systems Corp.). The temperature plate was mounted on a microscope stage (Eclipse TE-300; Nikon) and maintained at a constant 15°C. Aliquots of labeled sperm were diluted in ASW and transferred to an imaging chamber (final concentration ~$2 \times 10^5$ cells ml$^{-1}$). Epifluorescence images were collected with a Nikon Plan Fluor 40×/1.3NA oil-immersion objective using the Chroma filter set (ex HQ470/40x; DC 505DCXRU; em HQ510LP) and recorded on a DV887 iXon EMCCD Andor camera (Andor Bioimaging, NC). Stroboscopic fluorescence illumination was supplied by a Cyan LED no. LXHL-LE5C (Lumileds Lighting LLC, San Jose, USA) synchronized to the exposure output signal of the iXon camera (2 ms illumination per individual exposure). Images were collected with Andor iQ 1.8 software (Andor Bioimaging, NC) at 30.80 fps in full-chip mode (observation field of ~200×200 µm).

## 2.4. Image processing

The background fluorescence was removed by generating an average pixel intensity time-projection image from the first 94 frames (3 s) before uncaging, which was then subtracted from each frame of the image stack by using the Image calculator tool of ImageJ 1.49 u (*Schneider et al., 2017*). For *Figure 2d*, the maximum pixel intensity time projections were created every 3 s from background-subtracted images before and after the UV flash.

## 2.5. Quantitation of global changes of spermatozoa number and [Ca²⁺]ᵢ

To study the dynamics of overall sperm motility and $[Ca^{2+}]_i$ signals triggered by the distinct speract gradients we developed a segmentation algorithm that efficiently and automatically detects the head of every spermatozoa in every frame of a given video-microscopy archive (C/C++, OpenCV 2.4, Qt-creator 2.4.2). Fluorescence microscopy images generated as described previously were used. The following steps summarize the work-flow of the algorithm (*Figure 3—figure supplement 1*):

1. Segment regions of interest from background: This step consists of thresholding each image (frame) of the video to segment the zones of interest (remove noise and atypical values). Our strategy includes performing an automatic selection of a threshold value for each Gaussian blurred image ($I_G$) ($\sigma$ = 3.5 µm) considering the mean value ($M_I$) and the standard deviation ($SD_I$) of the image $I_G$. The threshold value is defined by: $T_I = M_I + 6SD_I$.
2. Compute the connected components: The connected components labeling is used to detect connected regions in the image (a digital continuous path exists between all pairs of points in the same component - the sperm heads). This heuristic consists of visiting each pixel of the image and creating exterior boundaries using pixel neighbors, accordingly to a specific type of connectivity.
3. Measure sperm head fluorescence. For each region of interest, identify the centroid in the fluorescence channel (sperm head) and measure the mean value.
4. Compute the relative positions of the sperm heads within the imaging field, and assign them to either R1, R2, R3 or R4 concentric regions around the centroid of the UV flash intensity distribution. The radii of R1, R2, R3 or R4, were 25, 50, 75 and 100 µm, respectively.
5. Repeat steps 1 to 4 in a frame-wise basis.

Step 1 of the algorithm filters out shot noise and atypical values; step two divides the images into N connected components for the position of the sperm heads; step three

quantitates sperm head fluorescence, and finally step four computes the relative sperm position on the imaging field. A similar approach has been recently used to identify replication centers of adenoviruses in fluorescence microscopy images (*Garcés et al., 2016*).

We automatically analyzed 267 videos of *S. purpuratus* spermatozoa, each containing tens of swimming cells, exposed to five distinct speract concentration gradients.

## 2.6. Analysis of speract induced Ca$^{2+}$ transients with immobilized spermatozoa

Imaging chambers were prepared by coating cover slips with 50 µg/ml poly-D-lysine, shaking off excess, and allowing to air-dry. Coated cover slips were then assembled into imaging chambers. Fluo-4 labeled spermatozoa were diluted 1:40 in ASW, immediately placed into the chambers, and left for 2 min, after which unattached sperm were removed by washing with ASW. The chambers were then filled with 0.5 ml of ASW containing 500 nM of caged speract and mounted in a TC-202 Bipolar temperature controller (Medical Systems Corp.). Images were collected with Andor iQ 1.7 software (Andor Bioimaging, NC) at 90 fps in full-chip mode, binning 4×4 (observation field of 200 µm x 200 µm). The imaging setup was the same as that used for swimming spermatozoa. The caged speract was photo-released with a 200 ms UV pulse delivered through an optical fiber (4 mm internal diameter) coupled to a Xenon UV lamp (UVICO, Rapp Opto Electronic). The optical fiber was mounted on a 'defocused' configuration to minimize the generation of UV light heterogeneities.

Images were processed off-line using ImageJ 1.45 s. Overlapping spermatozoa and any incompletely adhered cells, which moved during the experiment, were ignored. Fluorescence measurements in individual sperm were made by manually drawing a region of interest around the flagella with the polygon selections tool of ImageJ.

## 2.7. Sperm swimming behavior in different chemoattractant gradients

The sperm swimming behavior in response to a chemoattractant concentration gradient can be classified accordingly to their orientation angle ($\theta$), which is formed between their reference and velocity vectors (*Figure 4b*). For the sake of simplicity, chemotactic drifts (toward the source of the chemoattractant gradient) were considered to fall within the category of ($\theta < 60°$). The drift of swimming sperm in a direction perpendicular to the gradient results from orientation angles falling within the range $60° \leq \theta \leq 120°$. The instances of negative chemotactic drifts (opposite to the source of the chemoattractant gradient) were classified as those having higher orientation angles $\theta > 120°$ (*Figure 4—figure supplement 1*).

The proportion of spermatozoa orientated with low $\theta$ angles, i.e. toward the source of the chemoattractant concentration gradient is enriched in those gradients that give chemotactic responses: *f2 (p-value<0.001)* and *f5 (p-value=0.003)*; compare with *f1 (p-value=0.2)* and *f4 (p-value=0.51)*. Statistical comparisons were performed with the Pearson's Chi-squared test considering a probability of success of 1/3 for each type of response (non-responding cells were not considered).

The two tested negative controls for chemotaxis (Low Ca$^{2+}$ or High extracellular K$^{+}$ ([K$^{+}$]$_{e}$) for *f2* gradient) showed a complete distinct distribution that the corresponding *f2* gradient (*f2.0Ca p-value<0.001*, *f2.K p-value=0.01*, Fisher's exact test), i.e. as expected the proportion of cells experiencing chemotactic drift was significantly reduced on the negative controls.

Interestingly, the *f3* gradient provides the major stimulation of cell motility (the frequency of non-responsive cells drops down to ~2%), however in this experimental condition the proportion of cells responding toward the source of the chemoattractant gradient was not significantly distinct from the other two types of responses (p-value=0.12, Pearson's Chi-squared test).

In any tested gradient, the distributions of orientation angles have the same proportions between perpendicular and opposite to the source responses: *f1 (p-value=0.63)*, *f2 (p-value=1)*, *f3 (p-value=0.4)*, *f4 (p-value=0.84)* and *f5 (p-value=0.15)*. Statistical comparisons

4000

were performed with the exact binomial test considering a hypothesized probability of success of 0.5.

## 2.8. Spontaneous vs. speract-induced $[Ca^{2+}]_i$ oscillations

We characterized and compared the spontaneous vs. the speract-induced $Ca^{2+}$ oscillations (*Figure 3—figure supplement 3*) and conclude that they are completely different phenomena. Spontaneous $Ca^{2+}$ oscillations are only observed in about 10% of the analyzed population of spermatozoa (see Statistical analysis section in Materials and methods on the manuscript). Most of the time only one spontaneous oscillation is observed, and in the cases where more than one spontaneous oscillation is present (which accounts for ~20% of the spontaneous oscillations, i.e. only 2% of the total cells analyzed), they are significantly different in nature to the speract-induced $Ca^{2+}$ oscillations, judged as follows: they display a larger period and amplitude (~one order of magnitude) when compared to the speract induced oscillations (*Figure 3—figure supplement 3c* and *Figure 3—figure supplement 3d*). When these $Ca^{2+}$ spontaneous oscillations occur, if not very large, the cell will change direction randomly. If the oscillation is large enough, and is beyond a certain $[Ca^{2+}]_i$ threshold, the cell stops swimming altogether (see for example: *Wood et al., 2005*; *Guerrero et al., 2013*). After detection, we discarded cells undergoing spontaneous oscillations in the present work.

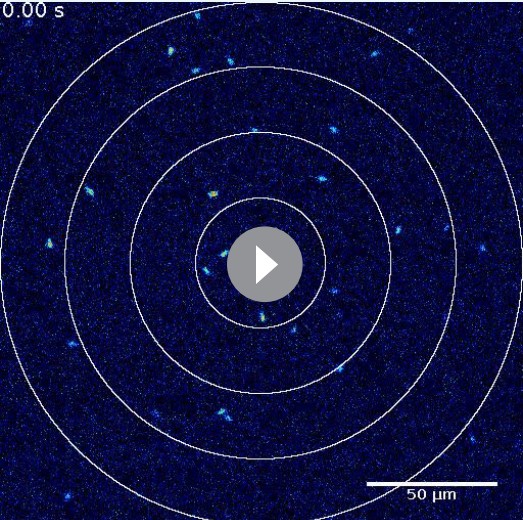

**Appendix 1—video 1.** Typical motility and $Ca^{2+}$ responses of *S. purpuratus* spermatozoa toward an *f3*-generated speract concentration gradient.     An optical fiber of 2 mm internal diameter (*f3*) was used for the UV light path to generate the speract concentration gradient. Other imaging conditions were set up as for *Video 1*. Note that spermatozoa located at R2, R3 and R4 regions prior to speract exposure swim up the concentration field toward the center of the gradient (R1). The pseudo-color scale represents the relative fluorescence of fluo-4, a $Ca^{2+}$ indicator, showing maximum (red) and minimum (blue) relative $[Ca^{2+}]_i$. Six *S. purpuratus* spermatozoa were manually tracked for visualization purposes. Scale bar of 50 μm.

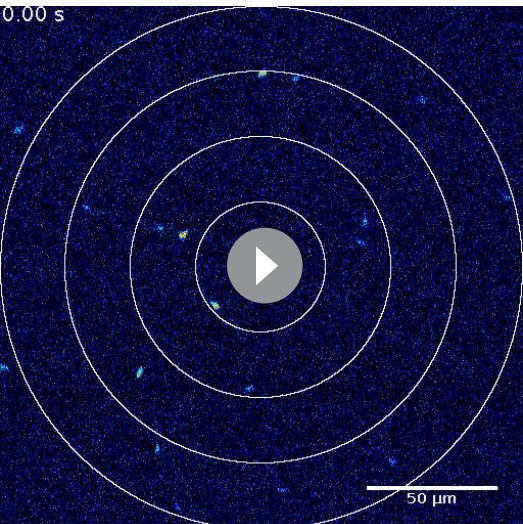

**Appendix 1—video 2.** Chemotaxis of *S. purpuratus* spermatozoa requires extracellular calcium. Spermatozoa swimming in artificial sea water with nominal calcium containing 10 nM caged speract 3 s before and 5 s after exposure to 200 ms UV light. Nominal calcium disrupts the electrochemical gradient required for $Ca^{2+}$ influx, hence blocking the triggering of the internal $Ca^{2+}$ oscillation by speract. The *f2* fiber (0.6 mm diameter) was used to uncage speract in this control. Other imaging conditions were set up as for *Video 1*. Note that spermatozoa re-located after speract uncaging but they failed to experience the $Ca^{2+}$-driven motility alteration triggered by speract. As a consequence, they failed to experience chemotaxis (compare with *Video 1*). The pseudo-color scale represents the relative fluorescence of fluo-4, a $Ca^{2+}$ indicator, showing maximum (red) and minimum (blue) relative $[Ca^{2+}]_i$. Six *S. purpuratus* spermatozoa were manually tracked for visualization purposes. Scale bar of 50 μm.

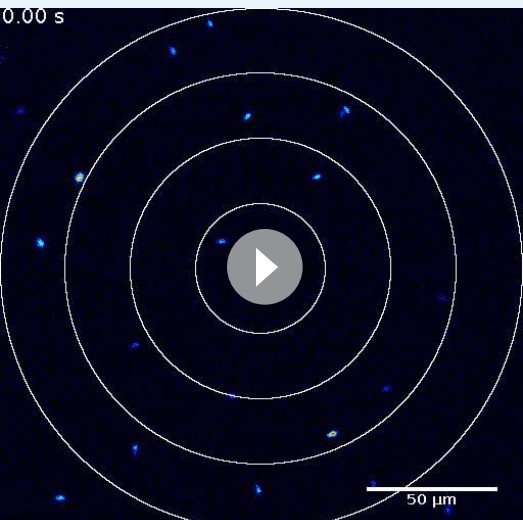

**Appendix 1—video 3.** Disrupting the $K^+$ electrochemical gradient blocks chemotaxis of *S. purpuratus* spermatozoa. Cells were swimming in artificial sea water containing 40 mM of KCl, and 10 nM caged speract 3 s before and 5 s after exposure to 200 ms UV light. High $K^+$ in the ASW blocks the membrane potential hyperpolarization required for opening $Ca^{2+}$ channels, and hence prevents the triggering of the internal $Ca^{2+}$ oscillator by speract exposure. The *f2* fiber (0.6 mm diameter) was used to uncage speract in this control. Other imaging conditions

were set up as for **Video 1**. Note that spermatozoa re-located after speract uncaging but they failed to experience the $Ca^{2+}$-driven motility alteration triggered by speract, and thus they failed to experience chemotaxis (compare with **Video 1**). The pseudo-color scale represents the relative fluorescence of fluo-4, a $Ca^{2+}$ indicator, showing maximum (red) and minimum (blue) relative $[Ca^{2+}]_i$. Six *S. purpuratus* spermatozoa were manually tracked for visualization purposes. Scale bar of 50 µm.

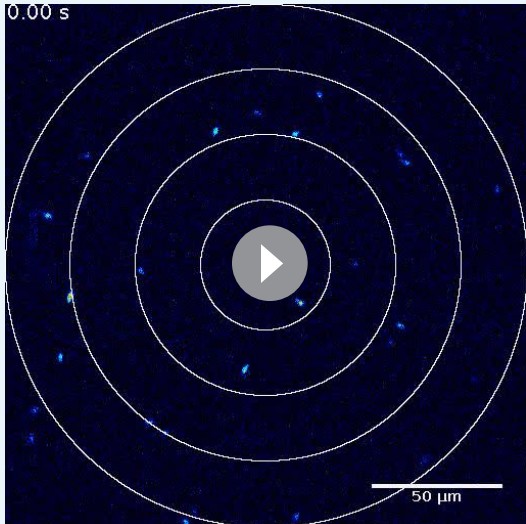

**Appendix 1—video 4.** Typical motility and $Ca^{2+}$ responses of *S. purpuratus* spermatozoa toward an *f1*-generated speract concentration gradient.     An optical fiber of 0.2 mm internal diameter (*f1*) was used for the UV light path to generate the speract concentration gradient. Other imaging conditions were set up as for **Video 1**. Note that some spermatozoa re-located after speract uncaging but they failed to experience chemotaxis (compare with **Video 1**). The pseudo-color scale represents the relative fluorescence of fluo-4, a $Ca^{2+}$ indicator, showing maximum (red) and minimum (blue) relative $[Ca^{2+}]_i$. Six *S. purpuratus* spermatozoa were manually tracked for visualization purposes. Scale bar of 50 µm.

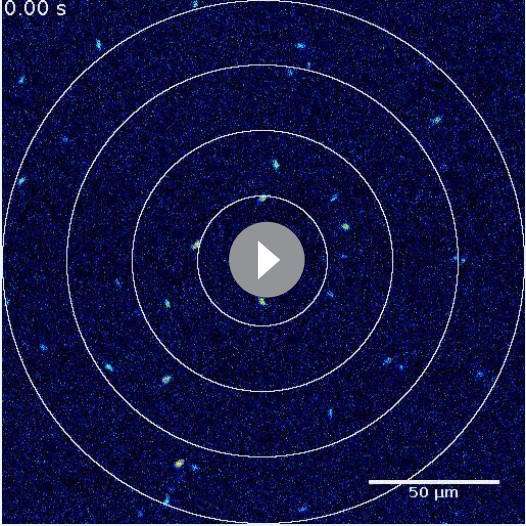

**Appendix 1—video 5.** Typical motility and $Ca^{2+}$ responses of *S. purpuratus* spermatozoa

toward an *f4*-generated speract concentration gradient.     An optical fiber of 4 mm internal diameter (*f4*) was used for the UV light path to generate the speract concentration gradient. Other imaging conditions were set up as for *Video 1*. Note that spermatozoa re-located after speract uncaging but they failed to experience chemotaxis (compare with *Video 1*). The pseudo-color scale represents the relative fluorescence of fluo-4, a $Ca^{2+}$ indicator, showing maximum (red) and minimum (blue) relative $[Ca^{2+}]_i$. Six *S. purpuratus* spermatozoa were manually tracked for visualization purposes. Scale bar of 50 μm.

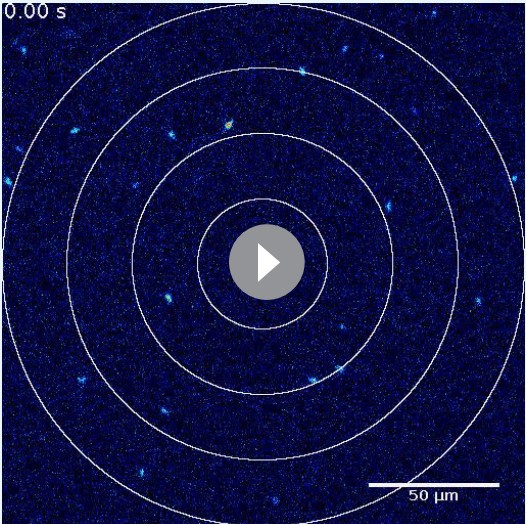

**Appendix 1—video 6.** Typical motility and $Ca^{2+}$ responses of *S. purpuratus* spermatozoa toward an *f5*-generated speract concentration gradient.     An optical fiber of 4 mm internal diameter (*f5*) was used for the UV light path to generate the speract concentration gradient. Other imaging conditions were set up as for *Video 1*. Note that spermatozoa located at R2, R3 and R4 regions prior to speract exposure swim up the speract concentration gradient, toward the center of the imaging field (R1). The pseudo-color scale represents the relative fluorescence of fluo-4, a $Ca^{2+}$ indicator, showing maximum (red) and minimum (blue) relative $[Ca^{2+}]_i$. Six *S. purpuratus* spermatozoa were manually tracked for visualization purposes. Scale bar of 50 μm.

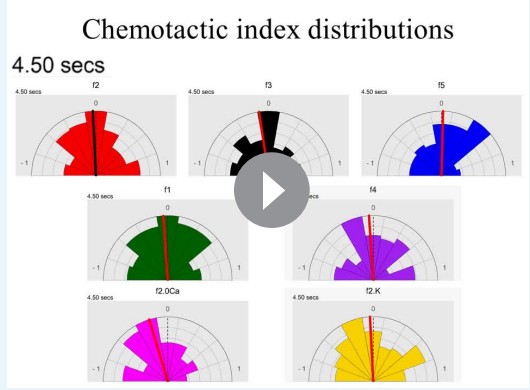

**Appendix 1—video 7.** Chemotactic index distributions.     Radial histograms of chemotactic indices from each different speract gradient. Black (*f2*) or red (rest) lines represent the median

of each distribution. This analysis was implemented from 4.5 s to 10 s. Speract uncaging was induced at 3 s.

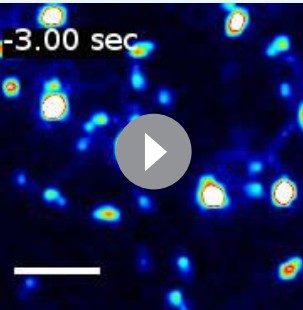

**Appendix 1—video 8.** Photo-release of caged speract induces $Ca^{2+}$ oscillations in immobilized *S. purpuratus* spermatozoa.     Spermatozoa were immobilized, by coating the cover slip with poly-D-lysine, in artificial sea water containing 500 nM caged speract, 3 s before and during 6 s after 200 ms of UV irradiation. The *f4* optical fiber was used for the UV light path to generate the speract concentration gradient. The optical fiber was mounted in a 'defocused' configuration to minimize the generation of UV light heterogeneities. 93 frames s$^{-1}$, 40x/ 1.3NA oil-immersion objective, 4×4 binning. The pseudo-color scale represents the relative fluorescence of fluo-4, a $Ca^{2+}$ indicator, showing maximum (red) and minimum (blue) relative $[Ca^{2+}]_i$. The brightness and contrast scale was adjusted for better visualization of $[Ca^{2+}]_i$ transients in the sperm flagella (as a consequence some heads look artificially oversaturated, however no fluorescence saturation was observed in the raw data).

