## [Decision Letter]

**Acceptance summary:**

The work of Ramirez-Gomez et al., is an important contribution to our understanding of sperm chemotaxis in sea urchins, a historically important class of organisms in the unravelling of this phenomenon. It was in these that sperm activating peptides such as speract were first identified; these play a role in triggering calcium increases that regulate dynein motor activity and thereby control motility. Through studies of various species of sea urchins and theoretical analysis of the limits of gradient detection the authors identify the boundaries for detecting chemotactic signals of *S. purpuratus* spermatozoa, and show that sperm chemotaxis arises only when sperm are exposed to sufficiently steep speract concentration gradients They show further that sperm chemotaxis arises through coupling between recruitment of speract molecules during sperm swimming and the internal Ca^2+^ oscillator.

**Decision letter after peer review:**

Thank you for sending your article entitled "Sperm chemotaxis is driven by the slope of the chemoattractant concentration field" for peer review at *eLife*. Your article is being evaluated by Naama Barkai as the Senior Editor, a Reviewing Editor, and three reviewers.

Given the list of essential revisions, the editors and reviewers invite you to respond within the next two weeks with an action plan and timetable for the completion of the additional work. We plan to share your responses with the reviewers and then issue a binding recommendation.

The reviewers had mixed opinions on this work, but reviewer #3 has raised a number of technical issues that need a clear response from you in order that we can reach a formal decision. Please pay particular attention to those items.

*Reviewer #2:*

There is nothing wrong with this paper. It gives a very thorough review of the well-researched field of chemotaxis including some solid modeling. The problem is that it isn't new or surprising (to me). The same manuscript was put online in June 2017 in bioRxiv with little note. The statement that "For almost three decades, chemotaxis had not been observed for the widely-studied *S. purpuratus* species under diverse experimental conditions, raising doubts about their chemotactic capabilities in response to the speract concentration gradients" is made without citation in both versions of the paper, but it doesn't seem to have been much of a mystery. The receptor density of low on *S. purpuratis,* which then requires a steeper gradient to overcome noise, as the authors have shown.

While there is nothing wrong here, it seems very academic and has previously attracted little attention, so I question why it should be published in *eLife*.

*Reviewer #3:*

I started the manuscript with excitement but it did not take long to recognise that the theoretical work has been not been implemented to sufficient standards of diligence; it appears unchecked for errors, both minor and fundamental, with examples of the latter including modelling assumption, equation solution and dimensions.

The limits of detection and the limits of when oscillators couple (e.g. that pendula on a wall are sufficiently coupled to synchronise) is interesting and is the concept in the context of chemotaxis explored here. However, thresholds by their nature are sensitive – the number of theoretical errors means that discussing and examining thresholds does not appear to be sound (as opposed to using controlled and justified approximations).

Hence, I am afraid I cannot recommend the manuscript for publication, with further details are below. I should note I have less confidence in the experimental aspects and leave this to other reviewers.

Equation (1):

- A list of assumptions should be provided with such equations. The authors have assumed a large Peclet number and it is not clear the Peclet number is large (it is, if the flagellum radius is used as the length scale, but the interaction of the fluid flow with the concentration field means such an assumption is not obviously valid). Appeal to Berg's paper is insufficient as the Peclet number is larger for bacteria, as the smaller the length scale the greater the effect of diffusion, and diffusion is dominant for an isolated bacterium. The assumption that Pe >> 1 is therefore a substantial one and should be justified – it is not clear, either way, whether it is true or false.

- The authors assume a spherical geometry for the flagellum. This is flawed. It is commented on in the SI and an alternative is given, but not used. I did not understand why an inappropriate and inaccurate approximation is used in the main text when the authors know this is an issue; no explanation seems to be present.

- The calculation of the receptor term, *N/[N + πa/s*], is for a spherical flagellum only and arises from interactions between the effects of adjacent receptors. For a fixed volume, as assumed, the sphere has minimal surface area and thus receptor interaction is highest given they are assumed placed at random. Thus, using the correct geometry with a fixed volume will reduce this receptor interaction effect and yet it is fundamental to the paper. Hence the authors are over-estimating the influence of one of the primary features they are testing for – this seems fundamental and one of the reasons I am recommending reject.

- *s* is the receptor effective radius not the chemoattractant radius (subsection “Species-specific differences in chemoattractant-receptor binding rates”).

Equation (5):

- This is also flawed. It is dimensionally incorrect as stated.

- To within a scaling to fix dimensions, it is the one-dimensional solution. Either the two-dimensional or three-dimensional solution is required here (depending on the gap between the cover slips, which does not appear to be given – my one comment on the experiments – the geometry should be clearly stated). The use of the incorrect spatial dimension changes the gradient term in the SDR expression and thus has major downstream impact, explicitly affecting what the authors are testing. Again, this seems fundamental in testing the chemoreception model and thus is a further reason I am recommending reject.

- Equation (5) does not respect the fact any solution without initial conditions imposed will satisfy invariance to shifts t -> t+q for any q and so cannot be correct (it needs t+t_0_ rather than t in the square root or t_0_ must be zero).

Equation (S4). Appendix 1 subsection “1.2. A condition for detecting a change in the chemoattractant concentration”. The chemoreception model uses the circumference of the sperm's circular path (v∆t) rather than its diameter, yet the diameter governs the range of concentrations the cell experiences. Given the ratio of circumference to diameter is π this constitutes a factor of π in the definition of *SNR* and feeds through to the rest of the paper. For studying thresholds, a factor of three can be very important and this contributes to my overall decision.

Appendix 1 subsection “1.1. On the estimate of maximal chemoattractant absorption” There are dimensional errors in the expression for the effective size of the binding site.

There are numerous points of presentation, only the more general are below as opposed to detailed minor points (e.g. no equation punctuation, use of ∂ for increments… which I have not documented).

- "Caution needs to be taken with the interpretations of the agreement of our data with such a generic model for coupled phase oscillators" – such a generic model does not pin down mechanism and an oscillator inheriting the frequency of its forcing does not seem unexpected, so I am also hesitant about what can be learnt from the interpretations. Similarly, for (over)statements in the manuscript e.g. "that spermatozoa exposed to steeper gradients experience lower uncertainty (i.e. higher *SNR*) to determine the direction of the source of the chemoattractant". Any theory with sensible monotonic relationships will show this trend so I am also not clear what is learnt from such observations. This is probably a point of presentation only, but I struggled on such points.

- It is unclear sometimes what is model prediction as statements that are derived from the models are often not stated as such making it harder to follow.

[Editors’ note: The editors accepted the authors’ plan for revisions asking for further expansion on certain points.]

3.2) Given the conviction of the authors, this should just be a simple case of providing an evidence-based estimate of the Peclet number for sperm (not bacteria – these are much smaller and thus unreliable for inferring the correct physical scales) to demonstrate transport is diffusively dominated. Such a demonstration is required.

3.3) Please evidence that is a legitimate approximation or use the expression for a cylinder. There is no demonstration that the difference between the two geometries does not change the presented results. Such evidence is required.

3.4) Please provide explicit evidence for your claims such that the use of this expression – based on a spherical flagellum – does not impact on the theoretical predictions and subsequent experimental comparisons, predictions, conclusions etc.

---

## [Author Response]

Reviewer #2:2.1) There is nothing wrong with this paper. It gives a very thorough review of the well-researched field of chemotaxis including some solid modeling. The problem is that it isn't new or surprising (to me). The same manuscript was put online in June 2017 in bioRxiv with little note.

Our work was uploaded in bioRxiv as part of the reviewing process in *eLife*. During these two years, and as a result of the positive interaction with the reviewers we have made significant improvements to our work. Since then, it raised the interest of several researchers of the community who provided personal feedback, enriching the process of revising the manuscript. The Abstract has been read 1,820 times, the HTML text downloaded 102 times, and the pdf 728 times. We therefore argue that these data are sufficient to demonstrate a wide interest in our findings. Furthermore, till this work, chemotaxis in *S. purpuratus* sperm has not been demonstrated and more importantly, an explanation of why it has not been observed under previously tested conditions was lacking (see detailed referenced answer below).

2.2) The statement that "For almost three decades, chemotaxis had not been observed for the widely-studied S. purpuratus species under diverse experimental conditions, raising doubts about their chemotactic capabilities in response to the speract concentration gradients" is made without citation in both versions of the paper, but it doesn't seem to have been much of a mystery. The receptor density of low on S. Purpuratis, which then requires a steeper gradient to overcome noise, as the authors have shown.

The reviewer seems to suggest that demonstrating sperm chemotaxis in *S. purpuratus* was a matter of time and is therefore trivial. In this regard, we counter by asking why has nobody found it before? To date, it has been more than three decades since the isolation of the founder member of the family of Sperm Activating Peptides (SAP) that regulate sperm motility, speract. Following the advice of this reviewer we will include references that state the observations made by some colleagues on the field of sperm motility:

Cook et al., 1994 cite: “Of the many identified echinoderm egg peptides (Suzuki and Yoshino, 1992), only resact is known to produce swimming responses (Ward et al., 1985), and a detailed understanding of signal transduction exists only for speract (Cook and Babcock, 1993a, 1993b)”.

In that article, the authors propose a model to explain the minimal molecular and cellular mechanism required for sperm chemotaxis, taking the current knowledge about speract signaling to extrapolate to the regulation of sperm motility during chemotaxis, while recognizing that speract was not demonstrated to be a chemoattractant itself. In the same manuscript the authors state:

“The changes in flagellar waveform and swimming paths produced by speract strongly indicate that a physiological role of this egg peptide is to modify sperm swimming behavior. Although chemotaxis to speract remains to be established, the similarities between the Ca^2+^ responses to speract and to the known chemoattractant peptide resact suggest that these represent part of a common mechanism for sperm chemotaxis”.

Solzin et al. (2004) compare the distinct signaling responses triggered by speract and resact on their con-specific spermatozoa, where similarities and differences between species are scrutinized, the corresponding section of the manuscript states: “speract, the peptide of *S. purpuratus*, does not display chemotactic activity (Cook et al., 1994). Despite this fact, the current model of sperm chemotaxis was readily generalized (Cook et al., 1994).

Moreover, in the same manuscript the lack of chemotactic responses of *S. purpuratus* to speract is stressed, citing the corresponding observations: “In a capillary assay, we also found no evidence for chemotactic activity of speract (unpublished data)”.

We note that the authors suggest that speract may act as a chemoattractant, but also leave open the possibility that it may serve functions other than chemotaxis. We did not and do not intend to provide here an exhaustive compendium of positive and negative results concerning assays tailored to explore the motility alterations driven by speract on *S. purpuratus* spermatozoa. Instead, we want to stress that the observed lack of chemotactic responses on these sperm cells has been recognized as a “mystery” in the field – if we aspire to generalize and interpret findings in sea urchin spermatozoa to chemotactic responses in other systems, then it is critical to accommodate and account for any apparent outliers, and not ignore them as inconveniently incongruent to the model.

Discussions about the molecular differences between species, that could explain the discrepancies between the motility responses triggered by either speract or resact, are present in the literature and remained unresolved until now. Quantitative dissimilarities on the number of receptors, swimming velocities, size of the flagellum, ecological reproductive niche, among others, have been noted and discussed in previous publications (reviewed in Guerrero et al., 2010). The present investigation is unique in the sense that it collects such information and, taking the seminal work provided by Berg about the physics of chemoreception as starting point (Berg and Purcell, 1977), and formulates a succinct but sufficiently complete model, which predicts the chemoattractant receptor density as an important quantity. Our findings indicate that the number of receptors determine the sensitivity that sperm have to reliably sample the egg positional information on a noisy background of signals, through measuring the slope of the chemoattractant concentration gradient. Our findings go much further, and are more generally relevant, than the singular observation of chemotaxis on *S. purpuratus.* They contribute to the general understanding of how information is transferred between the source and the searcher, when the channel of information is of chemical nature. The later becomes relevant when considering the ecological context where the chemical gradients are established, which for the case of marine species are shaped by the sliding of water bodies.

In summary, we present a rationale which establishes links that go from the molecular (number of receptors and coupling stimulation and signaling oscillators) to cellular (regulation of sperm motility) and ecological regimes (by understanding the scaling of the chemoattractant gradients as result of the hydrodynamical regime that shape them), where reproduction of marine invertebrates takes place.

2.3) While there is nothing wrong here, it seems very academic and has previously attracted little attention, so I question why it should be published in eLife.

This answer is a continuation of 2.1 and 2.2, which develop further explanations about the relevance of the present investigation. To complement this point we would like to recall the comment from reviewer #1: “In my opinion, the information content and quality of the figures seems now appropriate for *eLife*. Thus, overall, I think this joint experimental and theoretical work can be a valuable original contribution to the literature in the field of sperm chemotaxis”.

Reviewer #3:3.1) The limits of detection and the limits of when oscillators couple (e.g. that pendula on a wall are sufficiently coupled to synchronise) is interesting and is the concept in the context of chemotaxis explored here. However, thresholds by their nature are sensitive – the number of theoretical errors means that discussing and examining thresholds does not appear to be sound (as opposed to using controlled and justified approximations).Hence, I am afraid I cannot recommend the manuscript for publication, with further details are below. I should note I have less confidence in the experimental aspects and leave this to other reviewers.

We would like to start our reply with a general comment on the points raised by reviewer #3, summarizing the major points raised before discussing them one by one in dedicated answers.

Of major concern to the reviewer are several points that stem from the physical and mathematical grounds that sustain distinct aspects of the theoretical approaches used in our manuscript and, hence, of the validity of the statements derived from them. In what follows, we provide a discussion about the physical concepts used in our work, which are mostly based on the elaboration of minimal models whose approximations need to be stated and justified clearly. Unfortunately, these might have not been done to the required level of detail, leading to misunderstandings and erroneous interpretations. In particular, we emphasize that several of this referee’s comments appear to interpret our conclusions and findings as having the opposite meaning to that which we intended (i.e. 3.2, 3.3 and 3.9), hence, raising the doubts expressed in their comments. We are confident that the answers provided below will satisfy such inquiries.

Specifically, the major concerns were:

- Whether the geometric considerations that support the proposed chemoreception model justify our interpretations. In particular, the number density of speract receptors and their impact on the flux of chemoattractant (being or not within the linear regime).

- Clarify whether advective terms have to be taken into account for the elaboration of the chemoreception model.

- Whether the effective receptor size is computed correctly, and if not, whether it biases the observations gathered from the modeling approach.

- The use of an oversimplified diffusion model (1D) which might introduce errors in our estimate of the gradients explored by the sperm cells during chemotaxis. This, in turn, might impinge on the evaluated *SNR*, which is fundamental when testing the chemoreception model.

We show that the criticisms raised by the reviewer derive mostly from a misunderstanding of how we present concepts and approximations, and we hope that the current explanations lead the reviewer to revisit his/her appreciation about this manuscript, which currently departs from the appreciation of the other reviewers. For instance, reviewer #1 states that “the theoretical analysis […] and arguments presented by the authors in the revised version [is] convincing.” And reviewer #2 mention that this paper “gives a very thorough review of the well-researched field of chemotaxis including some solid modeling”. We hope that, in light of the provided answers, both the editor and the reviewers.

3.2) Equation (1):– A list of assumptions should be provided with such equations. The authors have assumed a large Peclet number and it is not clear the Peclet number is large (it is, if the flagellum radius is used as the length scale, but the interaction of the fluid flow with the concentration field means such an assumption is not obviously valid). Appeal to Berg's paper is insufficient as the Peclet number is larger for bacteria, as the smaller the length scale the greater the effect of diffusion, and diffusion is dominant for an isolated bacterium. The assumption that Pe >> 1 is therefore a substantial one and should be justified – it is not clear, either way, whether it is true or false.

The assumption behind Equation (1) is, in fact, quite the opposite to what the referee seems to imply here i.e. we are dealing with a purely diffusive flux into the spherical absorber. This is indeed the condition we should expect both for the individual bacterium discussed in Berg’s original work, and equally for the individual sperm cells discussed in the present work. The corresponding Peclet numbers remain small, i.e. diffusion dominates over advective transport, whether we use the flagellum radius or any other characteristic length scale for the swimming cell. This is a common scenario in the microbial world. To find high Pe (transport dominated by advection) at these micro scales we have to look, for instance, into collective behavior. That is something some of us have studied extensively in other contexts (e.g. Tuval et al., 2005) and which can, in practice, induce collective fluid flows at scales much larger than the individuals.

3.3) The authors assume a spherical geometry for the flagellum. This is flawed. It is commented on in the SI and an alternative is given, but not used. I did not understand why an inappropriate and inaccurate approximation is used in the main text when the authors know this is an issue; no explanation seems to be present.

We consider that using a spherical geometry per se is valid. It is a first order approximation that has been extremely useful and successful in the past in shedding light on a large number of problems with more complex geometries. This includes the first estimate of diffusive fluxes in this same chemotaxis problem (as Berg and Purcell showed in 1977). We have followed the same principle of "minimal modelling" that captures the main physics but that, at the same time, allows for simple characterization of the relevant parameters (e.g. the dependence with the number of receptors).

However, and for the sake of completeness, we have also included in the Supplementary information of our manuscript the more accurate computation of the absolute diffusive flux on a cylinder. This is a significantly more complicated expression than the compact solution for the spherical case. Nonetheless, and to a first order in an expansion in the slenderness of the flagellum, it is in fact approximately proportional (i.e. equal modulo a pre factor/proportionality constant) to the flux on a sphere. Hence, we strongly consider that the spherical approximation mostly used throughout the main manuscript to be neither "inappropriate" nor "inaccurate" as for comparing relative fluxes for which the aforementioned pre-factor/proportionality constant cancels out and, hence, becomes irrelevant. In the new version of the manuscript, we have now expanded the explanation for the used approximations and their validity as requested by the referee. We thank the reviewer for raising this concern.

3.4) The calculation of the receptor term, N/[N + πa/s], is for a spherical flagellum only and arises from interactions between the effects of adjacent receptors. For a fixed volume, as assumed, the sphere has minimal surface area and thus receptor interaction is highest given they are assumed placed at random. Thus, using the correct geometry with a fixed volume will reduce this receptor interaction effect and yet it is fundamental to the paper. Hence the authors are over-estimating the influence of one of the primary features they are testing for – this seems fundamental and one of the reasons I am recommending reject.

The receptor term, as originally deduced by Berg and Purcell, (1977), arises from the matching of two distinct limits: for a low number of receptors, the flux into independent patches leads to an overall diffusive flux into the sphere that is linear with the number of receptors. In the opposite limit of large surface coverage, the "interactions between the effects of adjacent receptors" leads indeed to the saturation of chemoreception. However, we believe "the primary features [we] are testing for" are the exact opposite to what the referee states: a low number of (non-interacting) receptors, sparsely covering the flagellum (i.e. with a large distance between receptors compared to receptor size) entails a non-saturated diffusive flux that, hence, depends on the number of receptors.

The cylindrical geometry, if anything, strengthens our assumption: the larger surface area of the cylinder gives a larger average distance between receptors and, hence, offsetting the saturation of the overall diffusive flux to higher receptor number. As a result, the flux strongly depends on the number of receptors, which is indeed one of the main points of our manuscript: it is incorrect to assume that the number of receptors is in fact large enough for all species for the perfect absorber approximation to be a valid one. Thus, by using the approximate spherical case, we are actually underestimating this effect, instead of overestimating as suggested by the reviewer. Such explanation has been now included in the manuscript.

3.5) s is the receptor effective radius not the chemoattractant radius (subsection “Species-specific differences in chemoattractant-receptor binding rates”).

Continuing from 3.4, our model derives from Berg’s original model of a sphere covered with N perfect absorber patches of radius *s*, the effective receptor radius. The dimensions of the speract receptor radius is not known, however Pichlo el al., (2014), provided an estimation of the radius of the resact receptor (the extracellular domain of the GC) of 2.65 nm. The value of *s* ~ 0.19 nm used in this work is about one order of magnitude smaller than such estimation. This value arises not from estimates of either receptor or chemoattractant sizes, but rather from an estimate of the effective size of the binding site, based on experimental measurements of chemoattractant binding kinetics (seeSupplementary file 1, and 3.9 below).

3.6) Equation (5):– This is also flawed. It is dimensionally incorrect as stated.– To within a scaling to fix dimensions, it is the one-dimensional solution. Either the two-dimensional or three-dimensional solution is required here (depending on the gap between the cover slips, which does not appear to be given – my one comment on the experiments – the geometry should be clearly stated). The use of the incorrect spatial dimension changes the gradient term in the SDR expression and thus has major downstream impact, explicitly affecting what the authors are testing. Again, this seems fundamental in testing the chemoreception model and thus is a further reason I am recommending reject.

We thank the reviewer for raising these observations, and we agree with both comments to a certain extent. Equation (5) is indeed dimensionally incorrect as stated, in the sense that there is a normalization constant/pre-factor that has not been explicitly included. It is, however, implicitly accounted for through the fitting procedure to the different UV profiles and, hence, this typo does not propagate any errors into the evaluated variables. We have now made this choice more explicit when first defining *c*_0_.

Equation (5) is indeed (modulo the aforementioned pre-factor) the solution to the 1D diffusion problem. Although this was done for the sake of simplicity, we agree with the reviewer in that using different spatial dimensions would have an effect on the estimated time-dependent gradients.

In nature, external fertilizers sperm cells tend to swim in spiral 3D trajectories. However, under the experimental conditions explored in this research, we are analyzing only the confined sperm swimming trajectories within the imaging chamber, swimming in 2D circular-like trajectories confined at a few microns above the coverslip. The UV flash that sets the initial chemoattractant distribution was focused at the imaging plane (~1-4 𝜇m above the coverslip). Hence, the correct diffusion problem indeed corresponds to that of a 2D diffusing regime.

We plan to shortly revisit our analyses within this 2D diffusive framework but, in the meantime, we provide here a first estimate for the induced errors. In Author response image 1, we compare the solution for the diffusive spreading between the 1D approximation and the 2D case. While the time evolution of the chemoattractant concentration profiles significantly differs (see Figure 1A and 1C), the relative gradients (i.e. locally normalized derivatives) are reasonably similar (Figure 1B and 1D). Remember that our proposed chemoreception model considers relative concentration gradients, instead of absolute ones. For a typical chemotactic sperm sampling chemoattractant concentration gradients over the course of a few seconds, the average error committed by assuming a 1D diffusive spreading (which would increase as a function of time as seen in Figure 1F) would be of the order of 25%. This error is significant and must be corrected for, which is the reason we plan to perform a recalculation prior to resubmission. However, we would like to stress that a 25% error in the estimated relative gradients will not modify substantially the main conclusions of our work, as these are based on an over 300% differences in experienced gradients between the different experimental conditions (i.e. different light profiles and different species sensitivities) as we have measured and presented in the manuscript.

**Author response image 1. respfig1:** Diffusion solutions in 1 dimension (1D) and 2 dimensions (2D). Diffusion simulation in 1D (**a,b**) and 2D (**c,d**) and the estimated relative error percentage between 1D and 2D (**f**). The numerical solution for the time dependent diffusion equation (i.e. the concentration profiles) for 1D and 2D, respectively, is shown in (**a**) and (**c**). The initial profile set at the time when the chemoattractant was activated with a UV flash (t = 3s) is highlighted as a blue dash line. Time evolution is shown in 1s intervals up to t = 10s (black lines). (b and d) depict the relative gradients (i.e. normalized derivatives) of the chemoattractant concentration. Black and red solid points indicate their spatial average at each time step, which in panel (**e**) is plotted as a function of time. In (**f**), the error percentage between 1D and 2D models was computed as (2D-1D)/1D from panel (**e**), and its temporal average is marked by a black solid point that indicates that the mean error between the 1D and 2D diffusion models for a typical chemotactic trajectory is about 25%.

3.7) Equation (5) does not respect the fact any solution without initial conditions imposed will satisfy invariance to shifts t-> t+q for any q and so cannot be correct (it needs t+t_0_ rather than t in the square root or t_0_ must be zero).

The referee is correct. This was actually a typo in Equation (5). Instead of “t”, one should have written “*t+t_0_”* as is present at a couple of lines below in the same paragraph (subsection “Computing the dynamics of speract concentration gradients”) when computing σ asσ=4D(t+t0). The value of *t_0_* was adjusted together with *c_0_*such that Equation (5) at *t=t_0_* accurately describes the shape of the UV light flash.

3.8) Equation (S4). Appendix 1 subsection “1.2. A condition for detecting a change in the chemoattractant concentration”. The chemoreception model uses the circumference of the sperm's circular path (v∆t) rather than its diameter, yet the diameter governs the range of concentrations the cell experiences. Given the ratio of circumference to diameter is π this constitutes a factor of π in the definition of SNR and feeds through to the rest of the paper. For studying thresholds, a factor of three can be very important and this contributes to my overall decision.

We fully agree with this comment, and believe the criticism stands from a simple misunderstanding: it is indeed the diameter of the swimming circle which "governs the range of concentrations the cell experiences". And this is the scale we have actually used to define our chemoreceptor estimates. This is specified, for instance, in Supplementary file 1 where we wrote “*v*, the mean linear speed of the spermatozoa, i.e. *Δr/Δt,* where *Δr* is the sampling distance (the diameter of the swimming circle).” We have now also made this choice of scale more explicit in the main text as well.

3.9) Appendix 1 subsection “1.1. On the estimate of maximal chemoattractant absorption” There are dimensional errors in the expression for the effective size of the binding site.

We understand the source of the confusion here and we apologize for not clarifying this estimate in more detail in the text. There are no dimensional errors in this expression. Following Phillips et al., "Physical Biology of the Cell", it is presented in a somewhat unusual manner: the estimated effective size is based on the affinity constant, *s*_e_ = *k*_on_/*D* where *k*_on_[s^-1^M^-1^s^-1^] and D[m^2^s^-1^], which requires first converting concentrations [M] into inverse volume [m^-3^]. This leads to the correct dimensions for s_e_[m] as discussed in the aforementioned textbook, and which is explicitly cited alongside this estimate in the Appendix.

3.10) There are numerous points of presentation, only the more general are below as opposed to detailed minor points (e.g. no equation punctuation, use of ∂ for increments… which I have not documented).

Thank you for raising this concern, the manuscript has been reviewed in search of punctuation errors and misuse of mathematical notation.

3.11) "Caution needs to be taken with the interpretations of the agreement of our data with such a generic model for coupled phase oscillators" – such a generic model does not pin down mechanism and an oscillator inheriting the frequency of its forcing does not seem unexpected, so I am also hesitant about what can be learnt from the interpretations.

We fully agree with the reviewer in that it is not that surprising to find matching of frequencies when dealing with two oscillators coupled through a forcing term. But we want to stress that the boundaries of the “region of synchrony” are by no means trivial. We recall that the experimental proof of the fact that the slope of the gradient is the driving force responsible for the oscillator coupling is a significant contribution of the present work. But what is most relevant to the former discussion is the existence of thresholds in the coupling strength, whose experimental calculations agree with our theoretical predictions based on the chemoreception model. In addition, such minimal model for coupled oscillators is also able to predict computed functional dependencies well documented in the literature, i.e. the observed temporal and frequency lags between the stimulation and signaling responses of the chemoattractant signaling pathway (Kaupp, 2003; Nishigaki et al., 2004; Bohemer et al., 2005; Wood et al., 2005, 2007; Strunker et al., 2006; Shiba et al., 2008; Guerrero et al., 2010; Alvarez et al., 2012, Pichlo et al., 2014).

3.12) Similarly, for (over)statements in the manuscript e.g. "that spermatozoa exposed to steeper gradients experience lower uncertainty (i.e. higher SNR) to determine the direction of the source of the chemoattractant". Any theory with sensible monotonic relationships will show this trend so I am also not clear what is learnt from such observations. This is probably a point of presentation only, but I struggled on such points.

Reviewer #3 is correct in the sense that we developed several arguments starting from a simple theory showing monotonic relationships, i.e. as the slope of the chemoattractant concentration gradient increases so does the *SNR*. However, the present research goes further than a theoretical proposal, as it develops alongside an experimental study that corroborates its predictions, i.e. sensing the presence of a chemoattractant gradient requires overcoming a boundary of detection at which the shaping of the decay length of the gradient, ‘the slope’, has a major role for sperm chemotaxis (for further discussion see responses to 2.3, 3.1 and 3.11).

3.13 It is unclear sometimes what is model prediction as statements that are derived from the models are often not stated as such making it harder to follow.

We revisited the whole manuscript to ensure the proper explanation of the cases where the statements originate either from the predictions of the model, or from the experimental observations.

[Editors’ note: The editors accepted the authors’ plan for revisions asking for further expansion on certain points. What follows is the authors’ additional responses to these points.]

3.2) Given the conviction of the authors, this should just be a simple case of providing an evidence-based estimate of the Peclet number for sperm (not bacteria – these are much smaller and thus unreliable for inferring the correct physical scales) to demonstrate transport is diffusively dominated. Such a demonstration is required.

Pe estimates the relative importance of advection (directed motion) and diffusion (random-like spreading) of "anything that moves". We are studying the motion of chemoattractant molecules: they are transported (relative to the swimming sperm cell) by its swimming while jiggling around by Brownian motion at the molecular scale. In this regard, there are distinct ways to conceptualize Pe, one for the chemoattractant molecules and one for the sperm cells themselves, which might explain discrepancies in Pe estimates, and hence interpretations. The Pe in our problem is small (<1), as we develop below. In contrast, Friedrich and Jülicher (2008) get to a more complex scenario where the Pe depends on the concentration gradient, taking values between 1 to 100. These two Pe are not the same, in the Friedrich and Jülicher approach, they are not looking at the thermal diffusion of chemoattractant molecules, but rather to an "effective" diffusivity used to characterize non-directed motion of the sperm cells themselves.

An evidence-based estimate of the Peclet number for chemoattractants can be provided by following the definition of the Peclet number Pe = UR/D, with the sperm swimming speed in the range U ~ [72-100 µm/s], diffusivity D ~ 240 µm^2^/s for the chemoattractant. The critical length scale R for the diffusive problem can be estimated by either i) computing the influx transport problem in a cylindrical geometry with the fluid flow parallel to the flagellar long axis (i.e. the sperm swimming direction) for which R is the flagellar width ~ 0.2 µm; or ii) for the simplified spherical cell approximation for which R is simply the equivalent spherical radius ae ~ [1.39-1.58 µm]. This renders Pe ~ [6e-2 – 6e-1] ≤ 1 for all experiments, which we have now included in Supplementary file 1 and which demonstrate that transport is diffusively dominated for an isolated single sperm cell (Acrivos and Taylor, 1962).

3.3) Please evidence that is a legitimate approximation or use the expression for a cylinder. There is no demonstration that the difference between the two geometries does not change the presented results. Such evidence is required.

As already mentioned in our previous reply, the use of the approximate spherical geometry is accurate when comparing relative fluxes (*J/J_max_*, see Figure 1—figure supplement 1 & Equation (A1)). The reason being Equations (A2) and (A3) in our Appendix 1 that show how the correct influx toward a cylinder, computed as a series expansion in slenderness, is approximately proportional (to a first order in the flagella slenderness) to the diffusive influx to an equivalent sphere; i.e. they are identical modulo a pre-factor. This pre-factor (~ 0.1 in our case) cancels out when computing relative fluxes (*J/J_max_*, see Figure 1—figure supplement 1 & Equation (A1)). Moreover, truncating out the second order term in the expansion (Equation (A2)) gives a < 5% error in these estimates. Hence, we stand by the simplest approximation when computing all our results.

3.4) Please provide explicit evidence for your claims such that the use of this expression – based on a spherical flagellum – does not impact on the theoretical predictions and subsequent experimental comparisons, predictions, conclusions etc.

The receptor term, as originally deduced by Berg and Purcell (1977), arises from the matching of two distinct limits: for a low number of receptors, the flux into independent patches leads to an overall diffusive flux into the sphere that is linear with the number of receptors. In the opposite limit of large surface coverage, the "interactions between the effects of adjacent receptors" leads indeed to the saturation of chemoreception (Figure 1—figure supplement 1). However, we believe that primary features we are testing for are the exact opposite to that which the referee states: a low number of (non-interacting) receptors, sparsely covering the flagellum (i.e. with a large distance between receptors compared to receptor size) entails a non-saturated diffusive flux that, hence, depends on the number of receptors (Figure 1—figure supplement 1). For a fixed volume, the cylindrical geometry strengthens our assumption: the larger surface area of the cylinder gives a larger average distance between receptors and, hence, offsetting the saturation of the overall diffusive flux to higher receptor number. As a result, the flux strongly depends on the number of receptors, which is indeed one of the main points of our manuscript: it is incorrect to assume that the number of receptors is in fact large enough for all species for the perfect absorber approximation to be a valid one (see Figure 1—figure supplement 1). The sphere is the geometric 3D figure of smaller area for a given volume. Thus, by using the approximate spherical case, we are actually underestimating this effect, instead of overestimating as suggested by the reviewer. Such explanation has been now included in the manuscript.

References:

Acrivos A, Taylor TD. 1962. Heat and mass transfer from single spheres in stokes flow. Phys Fluids 5:387–394. doi:10.1063/1.1706630